

# Seasonal and interannual (ENSO) climate variabilities and trends in the South China Sea over the last three decades

Violaine Piton [1, 2] and Thierry Delcroix [1]

[1] LEGOS, CNES, CNRS, IRD, UPS, University of Toulouse, France

[2] University of Sciences and Technology of Hanoi, Hanoi, Vietnam

Corresponding author: thierry.delcroix@legos.obs-mip.fr


**Abstract**
We present a short overview of the long-term mean and variability of five Essential Climate
Variables observed in the South China Sea over the last 3 decades, including sea surface
temperature (SST), sea level anomaly (SLA), precipitation (P), surface wind and water
discharge (WD) from the Mekong and Red Rivers. At the seasonal time scale, SST and SLAs
increase in the summer (up to 4.2°C and 14 cm, respectively), and P increases in the north.
The summer zonal and meridional winds reverse and intensify (mostly over the ocean), and
the WD shows positive anomalies. At the interannual time scale, each variable appears to be
correlated with El Niño Southern Oscillation (ENSO) indices. Eastern Pacific El Niño events
produce basin-wide SST warming (up to 1.4°C) with a 6-month lag. The SLAs fall basin-wide
(by up to 9 cm) during an El Niño event, with a 3-month lag. The zonal and meridional winds
strengthen (up to 4 m/s) in the north (weaken in the south) during all types of El Niño
events, with a 3-5-month lag. A rainfall deficit of approximately 30% of the mean occurs
during all types of El Niño phases. The Mekong River WD is reduced by 1/3 of the mean 7-8
months after all types of El Niño events. We also show increasing trends of SST as high as
0.24°C/decade and SLAs by 41 mm/decade. Increasing trends are observed for zonal wind,
which is possibly linked to the phase of the Pacific Decadal Oscillation, and decreasing trends
are observed for P in the north and both WD stations that were analyzed. The likely driving
mechanisms and some of the relationships between all observed anomalies are discussed.
**Keywords:** ENSO, trends, climate variability, essential climate variable, South China Sea,
seasonal variability




## 1. Introduction

The South China Seas (hereafter, SCS) is the largest marginal sea in Southeast Asia
and covers an area of 3.8 million km² from approximately 0-23°N and 99°E-121°E (Fig. 1).
This semi-closed basin is surrounded by South China, the Indochina Peninsula, Borneo and
the Philippines. It is open to the East China Sea through the Taiwan Strait, to the Pacific
through the Luzon Strait and to the Indian Ocean through the Malacca Strait and other
narrow straits. The SCS is the second busiest maritime route in the world; its topography is
rather complex with an average depth of 2000 m and maximum depths reaching 5000 m in
the northeast. There are wide and shallow continental shelves in the northwest and
southwest of the SCS and numerous islands such as Hainan, Paracel and Spratly.

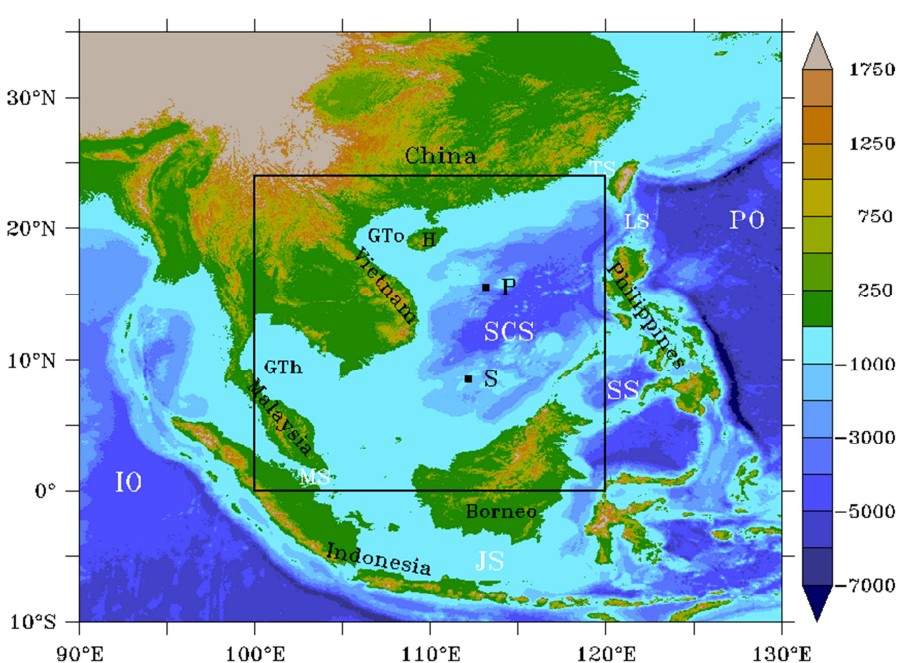

**Figure 1**. Bathymetry below sea level and topography above sea level (in m) of the South
China Sea region. Our research domain is enclosed with a black rectangle. The acronyms
denote, *GTh* Gulf of Thailand, *GTo* Gulf of Tonkin, *H* Hainan island, *IO* Indian Ocean, *JS* Java
Sea, *MS* Malacca Strait, *LS* Luzon Strait, *TS* Taiwan Strait, *P* Paracel Islands, *PO* Pacific Ocean,



*S* Spratly Islands, *SCS* South China Sea, and *SS* Sulu Sea. The bathymetry intervals are from 0
to 1000 m, and the topography intervals are from 0 to 250 m.

Located in the tropical Northern Hemisphere, the SCS region is influenced by both a

tropical and a subtropical climate, by the four adjacent monsoon subsystems, as well as by
the inflow from and the outflow to the surrounding oceans (see the reviews by Wang et al.,
2009, and Qu et al., 2009). At the annual time scale, the climate is mainly determined by two
alternative monsoon cycles, a southwest monsoon that brings wet and warm weather over
the area in the boreal summer (hereafter, all seasons will refer to the northern hemisphere
seasons) and a northeast monsoon that brings cold and dry conditions in the winter (Wyrtki,
1961; Wang and LinHo, 2002; Nguyen et al., 2014). Moreover, at the interannual time scale,
the SCS region is further influenced by the El Niño-Southern Oscillation (ENSO) phenomena.
Several studies have been conducted on the interannual (ENSO) variability of the SCS in
terms of sea surface temperature (Huynh et al., 2016; Liu et al., 2014; Tan et al., 2016; Yang
et al., 2015; Wang et al., 2006), sea level (Peng et al., 2013; Rong et al., 2007), surface winds
(Huynh et al., 2016), precipitation (Juneng and Tangang 2005; Nguyen et al., 2014; Räsänen
and Kummu, 2013; Räsänen et al., 2016), cyclone frequency occurrences (Camargo and
Sobel, 2005; Wu et al., 2005), and river runoff (Räsänen and Kummu, 2013; Xue et al., 2011).
Some of the aforementioned studies have focused further on the trends in the analyzed
variables, especially to estimate the potential impacts of global warming or decadal
variability (e.g., Peng et al., 2013; Xue et al., 2011).

As a complement to the large and growing body of regional climate-related

publications stemming from different authors, journal articles, datasets, and analyses of
specific time periods and time scales, the goal of the present study is to make an original
contribution to the literature by attempting to provide a concise and integrated analysis of
five key oceanic, atmospheric and terrestrial variables. To our knowledge, this study is
actually the first co-analysis of regional (SCS) oceanographic variables conducted over a



multi-decadal time frame made possible by the presence of long-term data products. For this
co-analysis, our approach is based on a coherent methodology, using the same technique for
each key variable, comparing each variable with the same ENSO indices, computing trends
over the same period, and using either new or longer datasets than the ones currently found
in the literature. Furthermore, this study provides an analysis of altimeter-derived Mekong
and Red Rivers discharge, which has not been previously documented, as well as in situ P
changes in near-coastal stations of Vietnam. The five key variables studied here are sea
surface temperature, sea level anomalies, surface winds, precipitation and river discharge,
all of which are considered an Essential Climate Variable (ECV) in the frame of the Global
Climate Observing Systems (GCOS, 2016). (In total, 50 variables covering three domains,
atmospheric, oceanic and terrestrial, are referenced as ECVs).

The five ECV datasets and the common data analysis methods based on temporal

filtering and EOF analysis are detailed in section 2. Their long-term means and standard
deviations (denoting the overall variability) are described in section 3 to set the context.
Then, their seasonal and interannual (ENSO) variabilities are analyzed in sections 4 and 5,
respectively. The long-term trends are finally documented in section 6. Aiming to be a short
overview paper, the knowledge gained from our study is compared to a variety of previous
results in all sections. A conclusion and discussion section is given in section 7.

**2.  Data and Methods**

Five ECVs are investigated: sea surface temperature (SST), sea level anomaly (SLA),

surface wind (SW), precipitation (P) and water discharge (WD). Among the multiple
available databases, all of the products that are described next have been selected based on a
compromise between several criteria: the dataset is as long as possible, grid resolution must
be as fine as possible, confidence, robustness and/or 'clear' documentation of the product
(partly based on a literature review).



SST. The primary datasets we selected are the 1° daily optimum interpolation SST
(daily OISST) version 2.0 developed by the National Oceanic and Atmospheric
Administration (NOAA; Reynolds et al., 2007; Reynolds 2009). (Note that all websites we
used to get the data are listed in the Acknowledgements section below). We extracted the
merged and gridded SST data covering the SCS from 0-24°N to 100-120°E and spanning
from 01/1982 to 12/2015 (34 years).
SLA. We used the multi-mission gridded sea level anomaly (MSLA) product produced
by AVISO+ (Archiving, Validation and Interpretation of Satellite Oceanographic Data) that
was based on TOPEX/Poseidon, Jason 1, ERS-1 and ERS-2 data. This product provides SLAs
relative to a 20-year mean from 1993 to 2012. This weekly data set is averaged to produce
the monthly mean data set in the present work and covers the same area as the SST data
from 01/1993 to 12/2015 (23 years) with a resolution of 0.25°x0.25°. The mean dynamic
topography (called the CNES-CLS13 MDT) was also obtained from AVISO+ and gives the
surface height above geoid over the period 1993-2012 (Rio et al., 2014). This product is
based on two years of GOCE (Gravity Field and Steady-State Ocean Circulation Explorer)
data, seven years of GRACE (Gravity Recovery and Climate Experiment) data and 20 years of
altimetry and in situ data.
SW. To investigate the SW, we selected the 10-m ERA-Interim merged and gridded
wind data developed by the European Centre for Medium-Range Weather Forecasts
(ECMWF; Dee et al., 2011). The data are available monthly on a 0.75°x0.75° spatial grid
covering the entire region for the period 01/1979 to 12/2015 (37 years).
P. The ERA-Interim reanalysis is also used as a source of precipitation data with the
same resolution and time coverage as previously noted. To further the study of precipitation,
we selected to investigate the land rainfall measurements provided by Vietnam's Hydro-
Meteorological Service. Among the 172 rainfall stations available, only 17 are considered in
this study. The selection was made regarding the following criteria: a) the data set must



cover at least 28 consecutive years, b) the dataset should not contain any missing data, c) the
spatial distribution must be homogeneous across Vietnam, and d) at least one station must
be located in each of the eight climatic zones of Vietnam described in Gobin (2016). The
locations and primary characteristics of the 17 selected rainfall stations are shown in Table
1. The correlation coefficients (R) between the in situ and ERA-Interim reanalysis time series
sampled near each station were high, and the values were similar for both seasonal and
interannual time scales (0.5<R<0.98). However, the ERA-Interim product tended to
underestimate the mean observed rainfall by approximately 16% (comparison made
between the 17 selected stations). Only 3 times series representing the northern (Lục Yên),
central (Đà Nẵng) and southern (Mỹ Tho) parts of Vietnam are presented in the following
section.

| Position | Name | Mean | Standard Deviation | Linear Trend |
|---|---|---|---|---|
| 21°.22'N, 103°.00'E | Điện Biên | 4.31 | 4.37 | $-2.4 \times 10^{-1}$ |
| 22°.06'N, 104°.43'E | Lục Yên | 8.18 | 6.05 | $-2.64 \times 10^{-1}$ |
| 21°.10'N, 105°.03'E | Minh Đài | 4.93 | 4.86 | $-7.8 \times 10^{-1}$ |
| 22°.50'N, 106°.31'E | Trùng Khánh | 4.90 | 4.38 | $-1.53 \times 10^{-1}$ |
| 21°.01'N, 107°.21'E | Cửa Ông | 4.97 | 5.73 | $-4.10 \times 10^{-2}$ |
| 20°.39'N, 106°.03'E | Hưng Yên | 3.94 | 3.80 | $7.42 \times 10^{-4}$ |
| 19°.10'N, 105°.38'E | Quỳnh Lưu | 4.37 | 4.49 | $-7.2 \times 10^{-2}$ |
| 18°.05'N, 106°.17'E | Kỳ Anh | 5.68 | 6.32 | $-4.71 \times 10^{-1}$ |
| 16°.80'N, 106°.60'E | Đông Hà | 10.48 | 11.55 | $-5.20 \times 10^{-1}$ |
| 16°.02'N, 108°.12'E | Đà Nẵng | 14.42 | 15.00 | 1.48 |
| 14°.46'N, 108°.44'E | Ba Tơ | 7.12 | 9.36 | $7.42 \times 10^{-1}$ |
| 13°.03'N, 108°.59'E | Sơn Hoà | 4.93 | 6.76 | $1.57 \times 10^{-1}$ |
| 11°.45'N, 108°.23'E | Liên Khương | 7.98 | 6.22 | $8.12 \times 10^{-1}$ |
| 11°.50'N, 106°.59'E | Phước Long | 6.78 | 5.78 | $3.53 \times 10^{-1}$ |
| 10°.47'N, 105°.56'E | Mộc Hoá | 4.05 | 4.14 | $-1.08 \times 10^{-1}$ |
| 10°.21'N, 106°.24'E | Mỹ Tho | 8.54 | 6.78 | $6.73 \times 10^{-1}$ |
| 9°.17'N, 105°.43'E | Bạc Liêu | 5.85 | 5.37 | $6.45 \times 10^{-4}$ |
| ST (21°.20'N, 105°.50'E) | Sơn Tây | 3540 | 2960 | -520 |



| CCV Satellites | Chroy Chang Var | 12400 | 8280 | -1220 |
| CCV in situ (12°.50'N, 105°.65'E) | Chroy Chang Var | 11600 | 10700 | -1340 |

**Table 1.** Means, standard deviations and trends from the 17 selected inland rainfall stations
in Vietnam (in mm/d and mm/decade), and the Red River ST, Mekong CCV in situ WD
stations and CCV from satellite measurements (in $m^3/s$ and $m^3/s$/decade), along with their
coordinates. The values were computed over the 1979-2006 period for the rainfall stations,
over the 1960-2010 period for the ST gauge station, over the 1960-2002 period for the in
situ measurements at CCV and over the 1996-2015 period for CCV from satellite
measurements.
WD. Finally, two river systems are considered in this study: the Red River and the
Mekong River, which are represented by the Son Tay (ST) station and the Chroy Chang Var
(CCV) station (located in Cambodia), respectively (see Fig 3e for locations). The latter station
was selected to avoid the tidal effects on the water level in the Mekong Delta. In situ data are
available at the ST and CCV stations, and water discharges were further computed from
altimetry at the CCV station. Radar altimetry discharges were obtained applying a rating
curve that related the water stage and discharge to the altimetry-based water level (e.g.,
Kouraev et al., 2004; Frappart et al., 2015). The time-series of altimetry-based water levels
was obtained using the Multi-Mission Altimetry Processing Software (MAPS – Frappart et al.,
2015) to process the along-track altimetry data from ERS-2 (1996-2003), ENVISAT (2002-
2010 on the nominal orbit) and Jason-2 (2008-2016 on the nominal orbit). The data, made
available by the Centre de Topographie des Océans et de l'Hydrosphère (CTOH), came from
the Geophysical Data Records (GDRs) D for Jason-2 and v2.1 for ENVISAT and from the
reprocessing of the ERS-2 radar echoes performed at CTOH to ensure the continuity with
ENVISAT for land studies (Frappart et al., 2016). In situ data at the ST station are provided
by Vietnam's Hydro-Meteorological Service and cover the period 1960 to 2010. The data
from the CCV station are from the Mekong River Commission and cover the period 1960 to



2002. The mean annual discharges from the gauge stations at the CCV station were
compared to the satellite data over the common period of measurements, 1996 to 2002. It
appeared that the annual means of the in situ measurements were adequately represented
by the satellite measurements since the slope of the regression line (between in situ and
satellite measurements) is close to one.
Climate indices. The interannual signal that was extracted from the original time
series (see the following paragraph) was compared to several well-known climate indices to
search for possible relationships with the atmospheric El Niño Southern Oscillation (ENSO)
Index (SOI) (Allan et al., 1996), the oceanic ENSO indices through the SST Niño3.4 (5°S-5°N,
170°W-120°W), SST Niño4 (5°S-5°N, 160°E-150°E) and SST Niño1+2 (10°S-0°, 90°W-80°W)
indices (Trenberth and Stepaniak, 2001 Rayner et al., 2003), and the El Niño Modoki index
(EMI) (Ashok et al., 2007). The SST Niño1+2 index is preferentially used to characterize
Eastern Pacific (EP) ENSO events. The SST Niño4 index is preferentially used to characterize
all types of El Niño events, but the authors could have chosen to use either the SOI or the SST
Niño3.4, as the phases of these three indices are almost equally sensitive to El Niño events
(Hanley et al., 2003).
Data processing. The climatological means, standard deviations (STDs), seasonal and
interannual variations were quantified for each ECV. The climatological means and STDs
were calculated over the entire length of each dataset (ranging from 23 to 37 years). The
seasonal variations were estimated by constructing a typical year for each ECV. This method
consists of computing the mean of each month (J, F, …, D) of the year over the multiyear
series. The interannual variations were estimated by subtracting the typical year from the
original series, which removes the mean seasonal cycle and filtering the remaining signal
with a 13-month low-pass Hanning filter. The filter passes almost no signals at periods less
than or equal to six months, filtering out the intraseasonal variability, and shortens each
time series by 6 months at the two extremities. Empirical orthogonal function (EOF)



analyses were then performed on the typical years and low-pass filtered time series, which
were first detrended. EOF analyses allow us to extract the main spatial modes of variability
and determine how they can change over time (e.g., Hannachi et al., 2007). Only statistically
consistent EOFs, based on North et al. (1982), are shown here, and all of the values
presented below are significant at the 90% confidence level (except for the P and WD
stations, as we did not perform EOFs on these series). Although statistically significant, some
EOF modes equal to or greater than two are not presented below because we did not find
any simple physical mechanisms possibly accounting for their related temporal and spatial
functions.

The long-term trends were computed by least-square fitting a straight line to the

monthly time series over the entire length of each dataset. The long-term trends over a
common period of measurement of 27 years (1979-2006) are also calculated and presented
when available (for P and SW from Era-Interim, P in situ and WD at ST).
**3.   ECV means and standard deviations**

The mean and standard deviation are presented for each oceanic and atmospheric

variable in Figures 2 and 3, respectively, and the atmospheric variables are available over
both the ocean and continent.

SST. The mean SST ranges from 24.5 to 29.5°C with the lowest values in the northern

part of the SCS basin and along the coasts of Vietnam (Fig. 2a), in agreement with Tuen
(1994), Chu et al. (1997) and Qu (2001) to name a few. Liu et al. (2004) also emphasized a
(relatively) cold tongue of water as the result of the advection of cold waters from the north
via the western part of the cyclonic gyre that develops in the winter in the area. The
standard deviation (Fig. 2b) ranges from 0.5 to 3.5°C with a high variability in the northern
part of the basin. On both the mean and standard deviation maps, the northeast-southwest
oriented isotherms are observed to have low temperatures and high standard deviations in
the north-northwest (see also Chu et al., 1997, and Qu, 2001). The lowest SSTs are mainly




caused by the northeast winds that blow in the winter, which first cool the seawater by
bringing cold and dry air and second generate a southwest-ward coastal current, which, in
return, brings cold coastal waters from the East China Sea into the SCS (Fang et al., 2006).

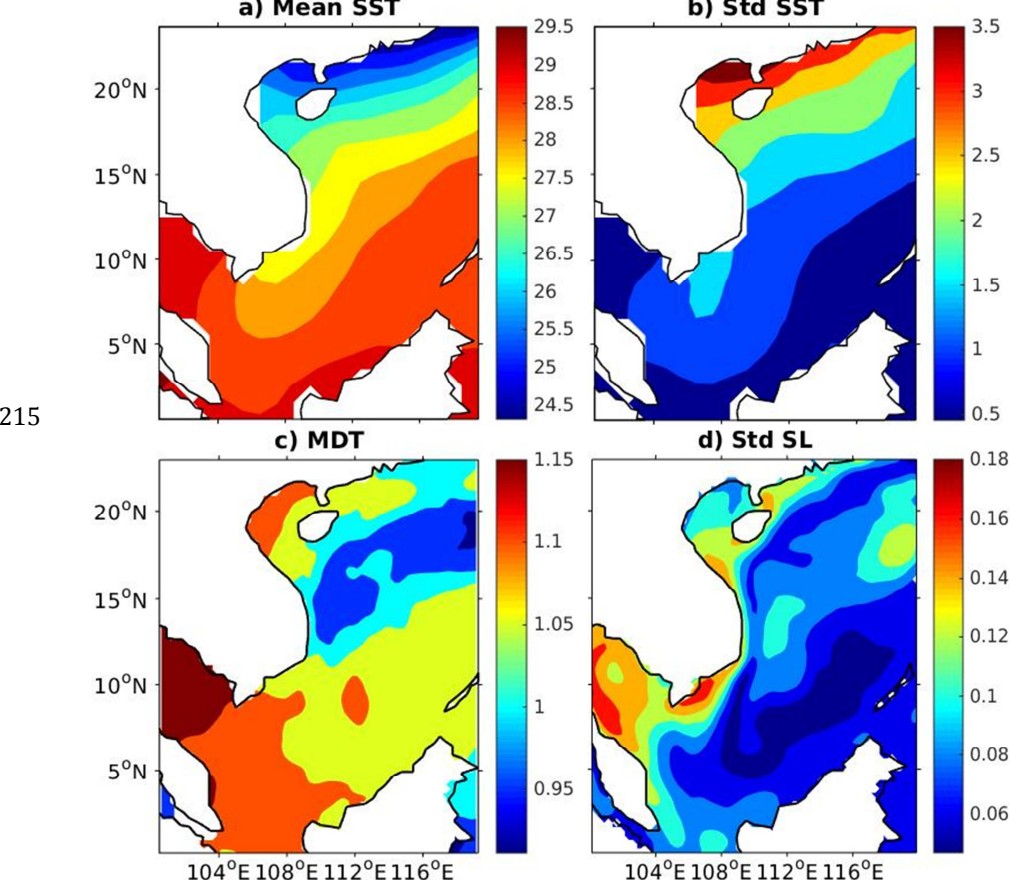


**Figure 2.** Mean and standard deviation of SST (a and b, respectively) in °C, and mean
dynamical topography (MDT) and standard deviation of SLA (c and d, respectively) in m. The
values were computed over 1982-2015 for SST, and 1993-2012 for MDT. The color codes
differ between the figures.

SLA. The map of the mean dynamic topography (MDT) shows spatial inhomogeneity

throughout the basin, featuring the mean surface geostrophic circulation (Fig. 2c). The
western and the southwestern parts of the SCS are characterized by the highest rates of



MDT, ranging from 1.05 to 1.15 m above the reference geoid. The lowest sea surface heights,
with values ranging from 0.9 to 1 m, stretch roughly from the southern tip of Vietnam to the
Luzon Strait, with minimum values near 18°N-118°E and 14°N-110°E. Zhuang et al. (2010)
found similarly low sea level heights at approximately 18°N-117°E northwest of Luzon
Island that extended southeastward to the east of Vietnam. Interestingly, the MDT resembles
the mean 0/400dbar dynamic topography derived from in situ temperature profiles and
mean TS curves (Qu, 2000). This similarity corroborates the existence of the cyclonic West
Luzon and East Vietnam eddies, which are two major features of the mean upper circulation
that are centered at the minimum values noted above. The SLA standard deviations range
from 6 to 18 cm (Fig. 2d), which is consistent with the results from Zhuang et al. (2010).
These results are comparable with previous results from TOPEX/Poseidon altimeter data
and Argo-tracked ocean surface drifter measurements presented by Ho et al. (2000).

SW. The mean surface zonal and meridional components are stronger over the sea

than over the land area, and the winds mostly blow W-SW (negative values in Fig. 3ac),
particularly in the northern part of the basin where the values are as high as 8.5 m/s
(approximately 16.5 knots). A mean southward flux is observed along the south of the
Vietnamese coastline. It appears that the mean surface winds and the standard deviations
associated with both zonal and meridional winds strongly depend upon the orographic
features (lower values above lands). The interpretation of the annual mean surface wind in a
region that is highly influenced by strong seasonal wind reversals due to the monsoon cycles
(see section 4) does not mean very much. The standard deviation values are the lowest
above land (from 0.5 to 4 m/s) and the highest above water (from 5 to 11 m/s) with cores of
maximum values near the southern tip of Vietnam. These values mostly reflect the seasonal
reversal (chiefly over the ocean) of the wind driven by the summer-winter monsoon cycles.




**Figure 3.** Means (a, c, e) and standard deviations (b, d, f) of the zonal and meridional surface

wind components in m/s (noted U and V, respectively) and precipitation in m/d (noted P).

Positive U and V wind components are directed to the East and the North, respectively. The





black dots on panel (e) denote, from north to south, the location of the inland rainfall
stations Lục Yên, Đà Nẵng and Mỹ Tho (see Table 1). The red dots denote the location, from
north to south, of the gauge stations at the Red River ST and Mekong CCV stations. The color
codes differ between the figures.
P. The south-central part of the SCS basin hosts the maximum P rate of approximately
8 mm/d. The lowest mean P rates are observed in the northern part of the basin and along
the Chinese coastline (Fig. 3e). As discussed in Wang et al. (2009), the sharp P contrast
between the NW and SE parts of the SCS arises from a similar SST contrast (see Fig. 2a). The
standard deviation of P ranges from 1.5 to 5.5 mm/d (Fig. 3f). The rainfalls data from in situ
measurements at 17 Vietnam stations (Table 1) range from 8.2 to 14.4 mm/d
(corresponding to approximately 3 to 5.3 m/yr). These measurements are consistent with
the results from Gobin et al. (2016) who noted that rainfall on land was measured within a
range of approximately 0.65 m/yr to more than 7.3 m/yr for the period 1960-2009. Here,
the mean rainfall for the in situ stations that were considered is approximately 3.8 m/yr
with the highest rate in the central part (Da Nang), as found in Gobin (2016). The standard
deviations range from 6.05 to 15 mm/d.
WD. At the ST Red River River station, the mean water discharge is equal to 3540
m³/s (for the period 1960 to 2010) with a standard deviation of approximately 2960 m³/s,
which is similar to the results from Vinh et al. (2014). At the CCV Mekong station, the mean
water discharge from the gauge stations and satellites appear to be consistent (11600 and
12400 m³/s, respectively).
**4. Seasonal variability**
SST. The first seasonal EOF mode on SST (94% of the total variance) represents an
annual cycle (Fig. 4b). The spatial function is positive over the entire basin and high in the
northwestern part of the basin and along the southeastern Vietnam coast (Fig. 4a). The
temporal mode reaches troughs in January-December and peaks in June-July, and they





correspond to the cold northeast winter monsoon and the warm southwest summer
monsoon, respectively. Thus, compared to the mean, this mode exhibits a cooling (up to -
4.8°C) over the entire basin in the winter and a warming (up to 4.2°C) mainly in the
northwest in the summer.
The second seasonal EOF mode (5% of the total variance) represents a semi-annual
cycle with peaks in April and November and troughs in January and August (Fig. 4d). The
spatial function is positive over the majority of the basin and negative along the Chinese and
northern Vietnamese coasts (Fig. 4c). In April, the SST can increase by as much as 1.5°C in
the Gulf of Thailand and south of Vietnam, and can decrease by -1.5°C in the Gulf of Tonkin.
As noted by Huynh et al. (2016), this semi-annual variability is mostly driven by oceanic
thermal advection along the northeast-southwest diagonal of the basin from two opposite
directions. These authors partially associate the spatial and temporal variabilities of the
second SST mode with the influence of an atmospheric anticyclone. In late March, the strong
development of this anticyclone weakens the northeast monsoon (winter monsoon), which
reduces the amount of clouds and rainfall over the SCS and allows more solar radiation to
reach the basin. As a result, the SST increases and reaches its peak in April and provides heat
and vapor for the onset of the southwest (summer) monsoon. The southwest winds trigger
the development of an anticyclone in the south and a weaker cyclone in the north, leading to
the advection of warm waters from the south to the northeast and cold waters from the
north to the southwest. In October, the summer monsoon dissipates and allows solar
radiation to warm the basin, reaching a peak in November.





**Figure 4.** Spatial functions (a, c, e) and associated temporal functions (b, d, f) of the first and

second (only for SST) EOF seasonal modes of SST and SLA. The products between spatial and



temporal functions denote anomalous SST (in °C) and SLA (in m) respective to the mean
values.

To further analyze the SST seasonal changes, typical years are represented in four

boxes, following the criteria of Qu (2001) based on mixed layer depths (boxes A to D in Fig.
5). A fifth box (E) was added to represent the region of high variability in the Gulf of Tonkin.
The results correspond to those of Qu and highlight the main influence of monsoon cycles on
SST in boxes A and E, while box C (and B and D to a lesser extent) also seems to be
influenced by the thermal advection described above.

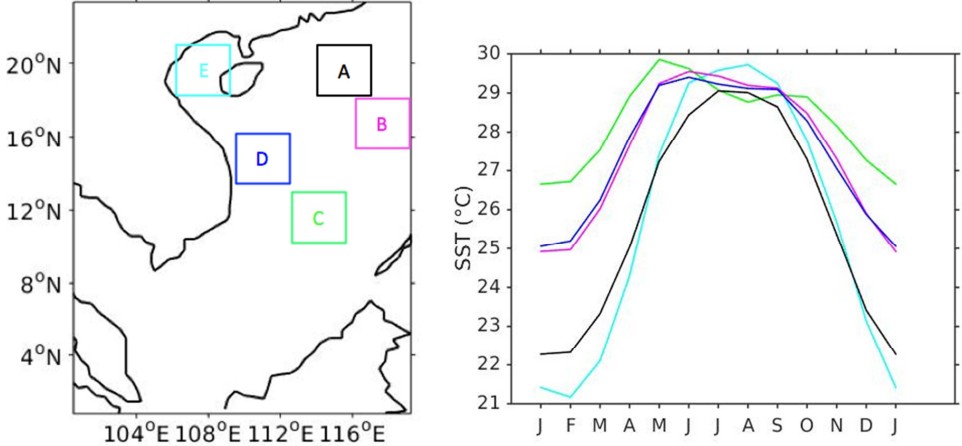

**Figure 5.** Locations of boxes (left) and time series of the relative SST monthly means (in °C)
over boxes A (black), B (magenta), C (green), D (blue) and E (cyan).

SLA. The first seasonal EOF mode on SLA accounts for 80% of the total variance of the

signal (Fig. 4ef). The spatial function shows positive values in the central and eastern parts
of the SCS and negative values in the western part of the basin. The temporal function
reveals higher-than-average (up to 0.14 m) SLAs in the summer in most of the basin and
lower-than-average SLAs along the coastlines of Vietnam, Cambodia and Malaysia. In the
winter, the situation reverses leading to negative anomalies over most of the basin and
positive anomalies along the coasts. Interestingly, this EOF mode also emphasizes lower-
than-average SLA in the winter near the center of the West Luzon eddy (as well as the East

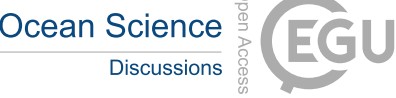

Vietnam eddy, although to a lesser extent). Logically, the timing of those seasonal SLA drops
corresponds to the maximum development of the two eddies inferred from hydrographic
observations (Qu, 2001). In addition, the seasonal sea level drops that are observed in the
summer along the coasts are consistent with the upwelling-favorable surface winds that
blow to the northwest at that time of the year (see below).

SW. The first seasonal EOF mode on surface wind anomalies contributes to up to 93%

of the variance for both zonal (U) and meridional (V) components. The two temporal
functions show an annual cycle, and the spatial functions show positive values over the
entire region and more intense values over the sea (Fig. 6a-d). In the summer, the eastward
(U) and northward (V) anomalies are then more intense (up to 17 m/s). The situation
reverses in the winter with negative anomalies up to 10 m/s. The January-February-March
(JFM) and June-July-August (JJA) mean wind vectors are shown in Figure 7, which stress the
seasonal monsoon reversal to ease the interpretation of the EOF analysis performed
separately on each wind component. The wind anomalies strongly depend upon the
orographic features of the region: the seasonal wind anomalies are stronger above the sea.







**Figure 6.** Spatial functions (a, c, e) and associated temporal functions (b, d, f) of the first

seasonal EOF modes on the U and V components of the surface wind, and P. The products

between the spatial and temporal functions denote anomalous U and V (in m/s) and P (in



m/d) respective to the mean values. The black dots in Figure 6e are the same as those in
Figure 3e.

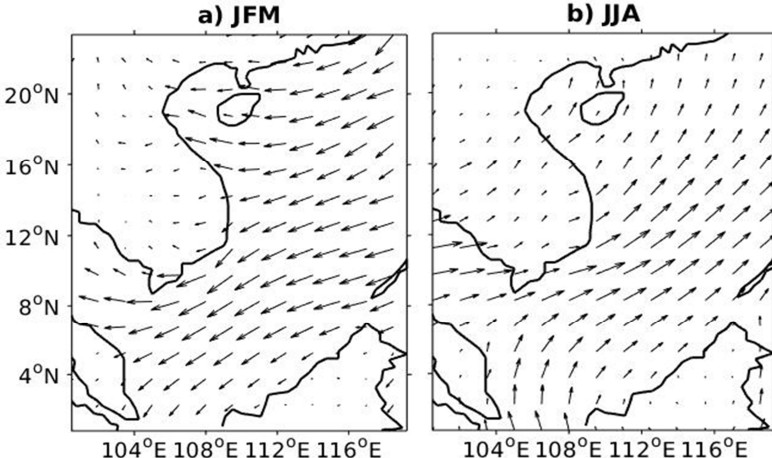

**Figure 7.** Mean surface wind in January-February-March (a) and June-July-August (b). The
arrows denote the wind vectors, and the longest arrows equal 8 m/s.

P. The first seasonal EOF mode on P contributes to 69% of the total variance (Fig.

6ef). A clear annual cycle appears on the temporal function. North of 8°N during the summer,
the entire region undergoes positive rainfall anomalies as intense as 6 mm/d, except along
the central coastline of Vietnam where the anomalies are slightly lower, which possibly
corresponds to the upwelling-favorable wind effects and relatively cold SST patterns near
the coast (Chu et al., 1997). During the winter, the signs of the anomalies reverse and the
region north of 8°N undergoes seasonal rainfall deficits compared to the mean.

The seasonal P anomalies relative to the means (see values in Table 1) of the three

selected in situ stations are shown in Figure 8. The anomalous values can be as high as 40
mm/d and as low as -8 mm/d, with a tendency to depict annual, Dirac-like, and semi-annual
functions at the Lục Yên, Đà Nẵng and Mỹ Tho stations, respectively. The results for the Lục
Yên station and the peak observed in the autumn at the Đà Nẵng station are qualitatively
similar to the observations (from different periods) by Nguyen et al. (2014). The authors



emphasize that the months of peak rainfall coincide with the southward migration of the
subtropical ridge and the intertropical convergence zone.

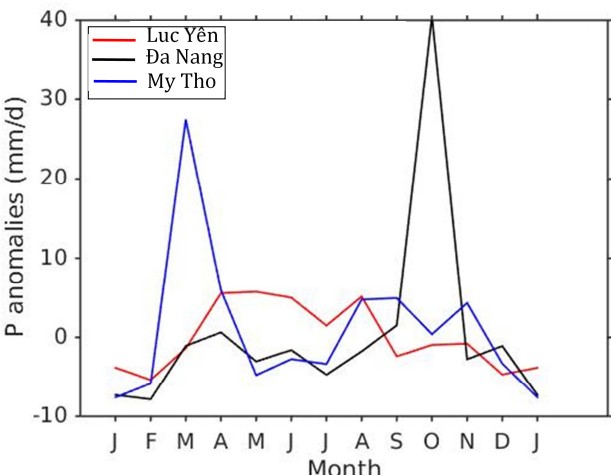


**Figure 8.** Seasonal variability of precipitation anomalies relative to the means from in situ
land stations, Lục Yên (red), Đà Nẵng (black) and Mỹ Tho (blue). The station locations are
denoted by black dots in Figures 3e and 6e (see also Table 1). The units are in mm/d.

WD. Figure 9 presents the seasonal variability relative to the means (see values in

Table 1) of the water discharge from the Red River (at ST) and Mekong River (at CCV). The
discharges observed at the ST station (red line) are much lower than those at the CCV station
(black and blue lines), with anomalies ranging from $-1.10^4$ to $+2.10^4\,\mathrm{m}^3/\mathrm{s}$. At the CCV station,
the highest discharge flux is observed in September (from both gauge station and satellite
measurements) with anomalies as high as $1.8\mathrm{x}10^4$ and $1\mathrm{x}10^4$ $\mathrm{m}^3/\mathrm{s}$ for the in situ and
satellite data, respectively. The entire water discharge series shows a peak in the summer
and a trough in the winter. This variability is probably driven by the monsoon dynamics
with a peak in water flow during the rainy season (summer monsoon) and negative
anomalies during the dry season (winter monsoon), as illustrated in Figures 6ef and
discussed above.

To explain the different amplitudes between the satellite and gauge station data at the

CCV Red River station (black and blue lines in Fig. 9), measurements over the common




period (1996-2002) were compared (not shown here). Both in situ and satellite
measurements showed similar amplitudes although the satellite measurements showed 1-2-
month lags in the peak and trough, for currently unknown reasons.

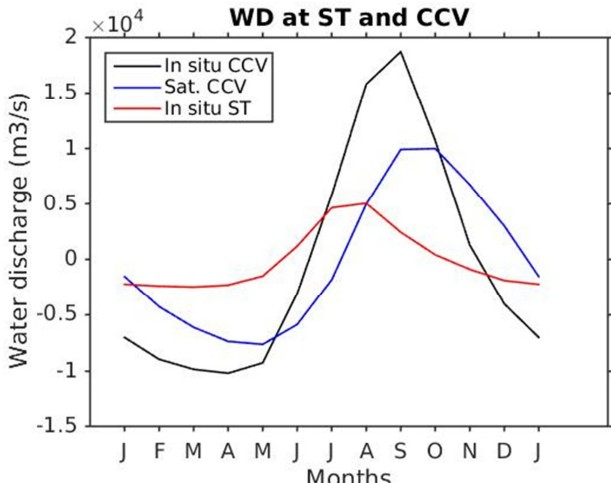


**Figure 9.** Seasonal variability of the water discharge (WD) anomalies relative to the means
at the (Red River) ST station from gauge station measurements (period: 1960-2010) in red,
and at the (Mekong River) CCV station from gauge station measurements (period: 1960-
2002, in black) and satellite measurements (period 1996-2015) in blue. The units are in $10^4$
$m^3$/s. The mean values are reported in Table 1.
**5. Interannual variability**

As noted in section 2, EOF analyses are performed to extract the main interannual

variability and compare with ENSO indices. The maximum correlation coefficients at given
lags between the interannual temporal function of the EOF and all (low-pass filtered) climate
indices are listed in Table 2.

| Time series | R (lag in months) | | | | |
|---|---|---|---|---|---|
| | Niño1+2 | Niño3.4 | Niño4 | SOI | EMI |
| SST EOF1 | 0.59(6) | 0.45(6) | -0.38(-12) | -0.44(-5) | -0.40(12) |
| SST EOF2 | 0.22(12) | 0.58(0) | -0.55(2) | 0.55(1) | -0.53(2) |
| SLA EOF 1 | 0.34(0) | 0.70(2) | 0.78(3) | -0.74(2) | 0.62(4) |
| SW U EOF1 | -0.51(3) | -0.69(4) | -0.64(5) | 0.65(5) | -0.42(7) |
| SW V EOF 1 | 0.60(0) | 0.74(2) | 0.58(3) | -0.71(3) | 0.28(4) |
| P EOF1 | 0.25(0) | 0.57(0) | 0.64(0) | -0.60(0) | 0.49(2) |



**Table 2.** Maximum correlation coefficients (R) at given lags (given in months in brackets)
between the ENSO indices and reported interannual EOF time functions (as shown in Figs.
11 and 13). A positive lag indicates that the indices lead the variable. All R values are
significant at the 90% confidence level.

_SST_. The spatial pattern of the first interannual EOF mode on SST (75.4% of the total

variance) is positive over the entire SCS with highest values along a northeast-southwest
diagonal (Fig. 10a). The associated temporal function (Fig. 10b) shows a maximum
correlation coefficient value of +0.59 with the Niño1+2 index (Table 2) with an index lead of
6 months. Thus, it appears that 6 months after the mature phase of an Eastern Pacific El Niño
event (represented by the Niño1+2 index, see section 2), a basin-wide SCS peak warming of
SST occurs. This peak is especially clear during the strong events of 1986-87 and 1997-98
when the SST increased by up to 0.7 and 1.0°C, respectively, in the central northern parts of
the SCS.

The spatial pattern, the anomalous amplitude and the lag are quite similar to those of

Chu et al. (1997) and Fang et al. (2006) that were obtained from different data sets and time
periods. They further resemble the averaged February and August double-peak structures of
the SCS SST composite anomalies derived from seven El Niño events covering 1950-2002 by
Wang et al. (2006). The lagging response of SST to ENSO events (observed here for Niño1+2,
and in Table 2 for Niño3.4, Niño4 and EMI) is described as a common feature in tropical
oceans. Klein et al. (1999) and Wang et al. (2004) attribute this lag to the effect of the
atmospheric bridge and the thermal inertia of the ocean mixed layer in the tropical Pacific.

The second interannual EOF mode on SST (7.3% of the variance) is represented by

positive spatial values in the SCS except near the Chinese and Vietnamese coasts and south
of Vietnam (Fig. 10c). The temporal function is best correlated to both the SOI and the Niño4
index (R= -0.55) with lags of one and two months, respectively, which are considered to be
representative of all types of ENSO events. Superimposed over the overall SST warming

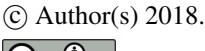



during the strong EP-type El Niño events depicted in the first EOF mode, it indicates that all
El Niño events produced SST decreases of approximately 0.1°C in the eastern half of the
basin and SST increases of up to 0.2°C along the coasts and south of Vietnam. The La Niña
events, not clearly visible in the first EOF mode (as in Wang et al., 2006), led to an SST
increase in most of the SCS, except along the coast and south of Vietnam where the
temperatures decreased. The first interannual EOF mode on the wind stress curl (not shown
here), an alias for Ekman pumping, represents 40% of the total variance, and the related
time function is correlated with the Niño4 index (R=0.6). Interestingly, its spatial pattern
proves rather consistent with the second interannual EOF mode on SST, being favorable to
upwelling (downwelling) in the eastern (western) half of the basin during El Niño (and vice
versa for La Niña).






**Figure 10.** Spatial functions (a, c, e) and associated temporal functions (b, d, f) of the first

and second interannual (only for SST) EOF modes on SST and SLA. The products between

the spatial and temporal functions denote anomalous SSTs (in °C) and SLAs (in m),





respective to the mean values. The Niño1+2 (b, in blue) and the Niño4 SST indices (d, f, in
red) are plotted over the time functions. The Niño4 index is reversed on panel (d). For
clarity, the temporal functions and the indices are filtered again with the H25 filter but are
described in the text with the TY-H13 filtration method.

SLA. Figure 10ef displays the first interannual EOF mode on the SLA, which accounts

for 48.1% of the total variance. The spatial patterns of this EOF mode are characterized by
the same negative sign throughout the basin (but with large horizontal gradients in
amplitude), which indicates that SLAs occur consistently over the entire SCS. The associated
temporal function is highly correlated (R=+0.78; lag=3 months) with the Niño4 index (Table
2). This EOF mode reveals that the sea level drop is largest three months after the mature
phase of an El Niño event. The negative SLA can be as strong as 5-10 cm in the eastern part
of the basin and along approximately 15°N. A reverse situation occurs during a La Niña year.
Rong et al. (2007) found similar spatial and temporal patterns for the period 1993-2005
with high correlation with the SOI (the only index tested). The observed anomalies are
consistent with the results presented by Fang et al. (2006) based on 1993-2003 satellite
altimetry data and with the results from a data assimilation model ran for the period 1993-
2002 (Wu and Chang, 2005).

SW. The first interannual EOF mode on zonal wind contributes to 64% of the total

variance of the signal (Fig. 11ab). The spatial function is positive in the southern part of the
region and negative north of approximately 14°N. The temporal function shows a high
negative correlation of R=-0.64 (Table 2) with the Niño4 index leading by five months (and
R=-0.69 with Niño3.4 leading by 4 months). This result indicates that 4-5 months after the
mature phase of an El Niño event, the eastward winds reach their maximal weakening (up to
-4 m/s) south of 14°N and get stronger (up to 4 m/s) north of 14°N. The situation reverses
during La Niña phases.




The first interannual EOF mode on meridional wind explains 49.4% of the total

variance with a positive spatial pattern north of approximately 6°N (Fig. 11c). The
corresponding temporal function is correlated with the SOI (R=-0.71) and the Niño3.4 index
(R=+0.74), with indices leading by 2-3 months. All types of El Niño events trigger an increase
in northward winds (e.g., up to 4 m/s during the 1997-98 event) north of 6°N and a decrease
of the same amplitude south of 6°N. These ENSO-related surface wind anomalies are
consistent with the results obtained by Wang et al. (2006) that were based on a composite
analysis, and those by Cheng et al. (2016) that were based on an EOF analysis, both obtained
from different wind field products and (shorter) time periods.




**Figure 11.** Spatial functions (a, c, e) and associated temporal functions (b, d, f) of the first

interannual EOF modes on the U and V wind components and P. The products between the

spatial and temporal functions denote anomalous winds (in m/s) and P (in mm/d),



respective to the mean values. The Niño4 SST index is plotted over the time functions. The
Niño4 index is reversed on panel (b). For clarity, the temporal functions and the indices are
filtered again with the H25 filter here but are described in the text with the TY-H13 filtration
method. The black dots in Figure 11e are the same as those in Figure 3e.

Figure 12 represents the composites of the anomalous wind vectors for the winter

during four strong El Niño events, noted JFM, and for the following summer, noted JJA. The
four El Niño events considered here are the 1982-83, 1986-87, 1991-92, and 1997-98
events, which were selected according to the studies of Chiodi and Harrison (2010, 2013).
The authors identified the abovementioned El Niño events as strong events that were
characterized by strong peaks in both Niño3.4 SST and outgoing long-wave radiation (OLR)
anomalies over the eastern-central equatorial Pacific. Figure 12a shows that the anomalous
winds were blowing N-NE, and when compared to the mean conditions in JFM when the
winds blow S-SW (Fig. 7a), this potentially leads to a weakened winter monsoon. The
anomalous JJA composites (Fig. 12b) show anomalous winds blowing S-SW, compared to the
mean conditions in JJA (when the winds blow N-NE (Fig. 7b)). Furthermore, each JFM and JJA
of each strong El Niño year were tested separately (not shown here), and the results showed
the same wind patterns as the JFM and JJA composites. Figure 12 compares the anomalies
over three months of a composite of strong El Niño events and is thus not directly
comparable with Figure 11abcd where the anomalies are presented relative to the long-term
mean, comprising all types of El Niño and La Niña events. Lastly, the same analyses were
performed on four La Niña years (1983-84, 1988-89, 1995-96, 1998-99). The surface winds
showed almost reversed patterns from the ones observed for El Niño years (not shown
here). The anomalous JFM composites for La Niña events showed anomalous wind blowing
S-SW, and the anomalous JJA composites showed anomalous wind blowing N-NE.



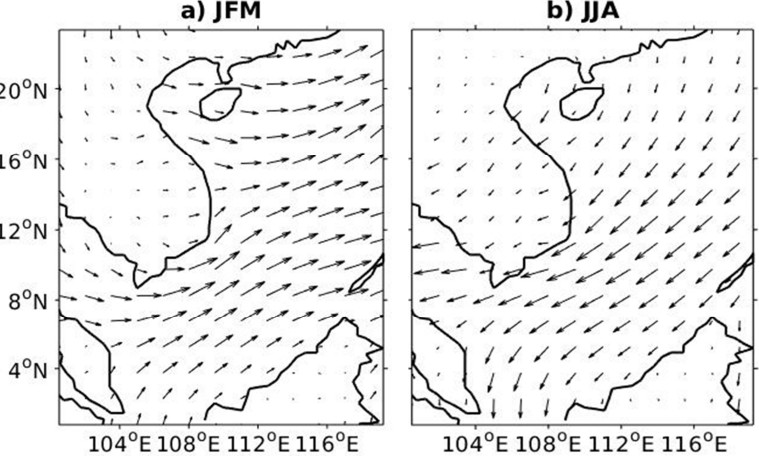


**Figure 12.** El Niño composites of the anomalous surface wind vectors in JFM (a) and JJA (b)
computed for 1983, 1987, 1992, and 1998, representing the 1982-83, 1986-87, 1991-92, and
1997-98 El Niño events, respectively. The arrows denote the vectors, with the longest
arrows equal to 1 m/s.

P. The spatial function of the first interannual EOF mode on P (48.4% of the total

variance) is negative over most of the region, except north of Vietnam, Laos and along the
Chinese coast, and it shows maximum values in the southeastern part of the region (Fig.
11e). The temporal function that is correlated with the Niño4 index (R=0.64 at 0 lag) reveals
that precipitation decreases during El Niño events and increases during La Niña phases. For
the case of the 1997-98 event, which was one of the strongest ENSO events ever recorded,
the amount of rainfall associated with EOF 1 (again representing only 48.4% of the variance)
reduces by as much as 1.6 mm/d in the southeast region, which is a reduction of 28%
relative to the mean value (Fig. 3f). The important P decreasing rate that was observed can
be explained by the eastward shift in convective activity in the equatorial western Pacific
during El Niño phases. Such shift leads to altered Walker circulation, generating a
subsidence area over the SCS (Wang et al., 2006).

From the 17 Vietnam rainfall stations tested, it appears that the linkage of

precipitation anomalies to ENSO is stronger at lower latitudes, with correlation coefficients





with Niño3.4 ranging from R=-0.19 to -0.45. As an example, at station Đà Nẵng where R=-
0.45, the rainfall decreases during El Niño phases by up to 4 mm/d during the 1997-98 El
Niño event (which is 70% of the mean). Nguyen (2014) also demonstrated that temperature
and rainfall variability are more strongly linked to ENSO events at lower latitudes of Vietnam
for the period 1971-2010. A study of the gridded precipitation data over the mainland of
Southeast Asia (1980-2013) that was undertaken by Räsänen et al. (2016) confirmed the
stronger influence of ENSO in the southern parts of the study area with high correlation to
ENSO during the spring of the decaying year.
WD. No obvious correlation was found between the interannual water discharge at
the ST Red River station and ENSO indices when the period 1960-2010 was considered.
However, when examining the period prior to the Hoa Binh Dam impoundment (from 1960
to 1978), the signal shows a statistically significant correlation to Niño4 (R=-0.47 at 12-
month lag). This suggests that the dam, by regulating the water flow according to the
dry/wet seasons and demand, may interfere with the interannual signal. At the CCV Mekong
station, both the 1960-2002 in situ and 1996-2002 satellite anomalies are correlated with
the SOI (+0.51 and +0.53, respectively, with an SOI lead of 6 months) and with the Niño4
index (-0.42 and -0.50 with 7-8-month lags, respectively). As an example, when considering
the 1997-98 event, the water discharge is reduced by approximately 4000 $m^3$/s (i.e., 34% of
the mean) approximately 7 to 8 months after the event. Xue et al. (2011) showed a similar
connection between ENSO and Mekong runoff (at Pakse, Laos, approximately 400 km
upstream of the CCV station) for the period from 1993-2005. It is important to consider the
WD data with caution as the precipitation anomalies over the entire Red River and Mekong
River watersheds (143,000 and 795,000 $km^2$, respectively) are integrated and may relate
processes that occur outside of the area considered here and potentially far away from any
ENSO influence.
**6. Trends**





Trends are calculated over the total length of each dataset. Since the series differ in
lengths, they are also calculated over a common 27-year period (1979-2006) for most of the
datasets (P in situ, P and SW products from ERA-interim, WD at ST). No significant
differences appeared between the trends that were calculated over the entire length of the
dataset and over the common period (both values are discussed when possible).
SST. The 1982-2015 trend shows an increasing SST trend with the highest values in
the northeastern part of the basin and maximum values west of the Luzon Strait and along
the Borneo coastlines of approximately 0.24°C/decade (or a total rise of 0.82°C of for the
period 1982-2015). A small SST decrease is observed along a very narrow coastal band off
central Vietnam and in the northern part of the Gulf of Tonkin of about -0.06°C per decade
(Fig. 13a). The computation of the wind stress curl trend (not shown here), an alias for
Ekman pumping trend, proves to be inconsistent with the spatial variations and even with
the sign of the SST trends. Fang et al. (2006) also found higher warming rates in the northern
deep basin and lower rates in the southern SCS with an area-mean SCS SST trend of
approximately 0.26°C/decade (from 1982 to 2004). They further identified a period of faster
warming rates between 1993 and 2003 with a rate up to 0.50±26°C/decade. Based on the
observed and simulated SCS SST from 1977 to 2013, Mei and Xie (2016) found similar rising
rates of SST of approximately 0.18 to 0.25°C/decade. For the 1960-1990 period, Casey and
Cornillon (2001) calculated a global ocean mean rate of 0.14±0.04°C/decade. Assuming that
the recent (although different) three-decade trends can be compared, our results indicate
that the 1982-2015 SCS warming rate is approximately 1.7 times greater than the 1960-
1990 global mean rate of Casey and Cornillon (2001), and approximately 2.2 times greater
than the 1971-2010 global mean rate discussed by the IPCC (2014).
As a cautionary note, it is important to keep in mind that large discrepancies exist
between the published estimates of 21st century SST trends, especially in the tropical Indo-
Pacific. These discrepancies seem to partly result from varying estimations of ENSO





influence in the computation of long-term trends, as discussed in Solomon and Newman
(2012). The influence of ENSO on the trend computation is illustrated in the last paragraph
of this section.








**Figure 13.** Trends of SST, SLA, surface wind components (U and V) and vectors, and P (a, b,
c, d, e, f, respectively) in °C/decade, mm/decade, m/s/decade and m/d/decade, respectively.



The arrows denote the wind vectors, and the longest arrows are equal to 1x10⁻³m/s/decade.
The color codes differ between the figures. The trends in the black square of Figure 13a are
discussed in the main text.

SLA. The map of the sea level trends for 1993-2015 shows an increasing trend

throughout the SCS basin with high spatial variability from 3 to 70 mm/decade and a mean
rising rate of approximately 41 mm/decade (Fig. 13b). A tongue of rapidly increasing rates is
observed from 14 to 20°N and along the longitudes 110 to 116°E, corresponding to the
location of the minimum of MDT (observed in Fig. 2c). This finding suggests that the sea level
must be rising in the low center of the cyclonic loop current that is generated by the
intrusion of a branch of the Kuroshio Current through the Luzon Strait (Farris and Wimbush,
1996; Ho et al., 2000), which results in a slowdown of the related Luzon Strait and East
Vietnam cyclonic eddies. Peng et al. (2013) found similar sea level rise rates of
approximately 39 mm/decade for the period 1993-2009 using the same data product. These
results are, as well as the results found here, faster than the global mean rate of sea level rise
of 28±0.4 mm/decade (obtained for the period 1993-2003 with the TOPEX/Poseidon
altimeter) (Cazenave and Nerem, 2004) and the global mean sea level for 1993-2015 of
approximately 3.3 mm/decade that was recently obtained by Dieng et al. (2017). In line with
the analysis of Meyssignac et al. (2012), Cheng et al. (2016) indicated that the PDO
contributed to 72% of the total sea-level rise in the SCS during the 1993-2012 period. The
authors suggested that the intensification of the easterly winds associated with the PDO in
the last two decades leads to the increase in the steric sea level by deepening the
thermocline in the Western Tropical Pacific. Finally, they indicated that the remaining 28%
of the sea level trend corresponded to the global sea-level rising rate (of 18±0.3
mm/decade).

SW. The trends of the two wind components (Fig. 13cd) range from -0.48 to +0.18

m/s/decade, respectively, with an increase of the zonal component (U) for the period 1970-



2015 in the northern and southern parts of the region and along the Vietnamese coastline
(reduced westward winds). Over the common period of measurements from 1979 to 2006,
the SW trends range from -4.3 to 1.7 $10^{-2}$ m/s/yr. The meridional component seems to
mainly weaken over the region, except in the eastern part of the SCS and above Borneo.
These results are consistent with the results from Fang (2006), who found an area-mean
trend of the zonal component of 0.56±35 m/s/decade for the period 1993-2003. However,
we did not find an obvious trend for the meridional component. Part (not estimated here) of
the observed trend in the zonal wind is directly linked to the phase of the PDO, as suggested
by England et al. (2014) and others, since trade winds have considerably strengthened over
the past two decades. Figure 13e summarizes Figures 13cd in a vector form and shows that
the maximum rate of the trend is $1 \times 10^{-2}$ m/s/decade, which occurs toward the south along
the Vietnamese coastlines and in the Gulf of Thailand.

P. The map of the linear trend of P clearly shows a rainfall deficit north of 8°N of

approximately 6 mm/d/decade over the period considered (1979-2015), with maximum
values inland (Fig. 13e). Over the common period of measurements from 1979 to 2006, the
decreasing trend is approximately -6.8 mm/d/decade. The Lục Yên station also shows a
decreasing rainfall trend ($-2.64 \times 10^{-1}$ mm/d/decade). However, the Đà Nẵng and Mỹ Tho
stations both show increasing precipitation trends with rates of $1.48 \times 10^{-1}$ and $6.73 \times 10^{-1}$
mm/d/decade, respectively. So far, we are not able to determine the origin of the observed
discrepancies between the satellite and in situ measurements (as well as the discrepancies
about the mean, as mentioned in section 2). Comparison with other satellite products is
recommended here. For the period 1961 to 1998, Manton et al. (2001) also showed that the
number of rain days over land (at least 2 mm of rain) has significantly decreased throughout
Southeast Asia. They associated this decrease with the predominance of El Niño events since
the mid-1970s (Trenberth and Hoar, 1997).





WD. All of the stations show small decreasing trends in water discharge. At the ST
Red River station, the water flux declined by -520 m²/s/decade (representing 1.5% of the
mean water discharge) over the period of 1960-2010. For the common period of
measurement (1979 to 2006), the water flux declined by approximately -420 m²/s/decade.
This decreasing trend has also been emphasized by Vinh et al. (2014); it is partially due to
the impoundment of the Hoa Binh Dam in the late 1980s, which has decreased the water
flow since then. The trends at the CCV Mekong station also showed decreasing rates of -1220
m³/s/decade (0.98% of the mean WD) and -1340 m³/s/decade (0.87% of the mean WD) for
satellite and in situ data, respectively. Lu and Siew (2006) investigated the disruption in
water discharge at stations on the Lower Mekong River that was induced by the cascade
dams in the upper part of the main stream of the Mekong River and found a declining trend
during the dry season. Furthermore, Xue et al. (2011) showed that the runoff of the lower
Mekong River was more closely connected to precipitation and ENSO in the post-dam period
(1993-2005) than in the pre-dam period (1950-1993). Distinguishing and isolating an effect
from another one (precipitation, water use and water regulation of a dam, ENSO) on water
discharge would provide crucial information on the behavior and possible forecasting of a
trend.
ENSO and long-term trend. The study periods and time series lengths are responsible
for not only the slight trend discrepancies observed between the literature and our results
found for each ECV but also the way the timing and the amplitude of ENSO may affect the
computation of long-term trends. To illustrate this issue, the 25-month Hanning filtered SST
and the corresponding linear trends were computed over the box 14.5-19.5°N, 112.5-
117.5°E roughly centered in the middle of the northern SCS (see the black square in Fig.
13a). Looking at Figure 14, the large SST anomalies that are characteristic of ENSO events
(see the ENSO indices in Fig. 10ab) clearly impact the computation of the linear trends. To
crudely estimate the expected effects, the 1987-1988 and 1997-1998 El Niño periods were



then removed, and the linear trends were recomputed. Ignoring these four El Niño years
increased the SST trends by as much as 1.4°C/decade (from 1.06 $10^{-1}$ °C/decade to 1.53 $10^{-1}$
°C/decade).

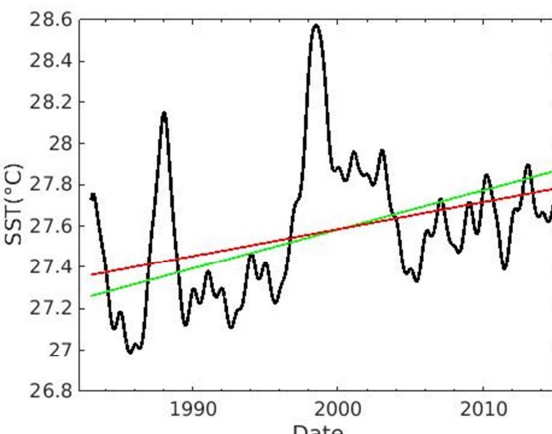


**Figure 14.** Time series of the 25-month Hanning filtered SST averaged over the box 14.5-
19.5°N, 112.5-117.5°E (in black), the corresponding linear trend (in red) and the linear trend
corresponding to the series shortened by four El Niño years (1987-88 and 1997-98) (in
green). The location of the box is denoted by the black square in Figure 13a.

**7. Conclusion and discussion**

In this study, we successively analyze the long-term means, standard deviations,

seasonal and interannual (mostly ENSO-related) variability and long-term trends of five key
ECVs (sea surface temperature, sea level anomaly, surface wind, precipitation, and water
discharge) to further document, validate, corroborate and/or complete the current scientific
knowledge on the climate variability in the South China Sea.

The analysis of the first (or first two) EOF mode(s) on each ECV clearly exhibits the

seasonal variability of the SCS under the influence of monsoon. In the summer, the SST is
found to increase throughout the basin, the SLA increases in the central and eastern parts of
the basin, positive rainfall anomalies are observed north of 8°N, the eastward and northward





wind components are stronger (reflecting the seasonal monsoon reversal) and the water
discharge increases.
A summary of the ENSO-related changes and long-term trends of the five ECVs
analyzed is provided in Table 3.

|  | SST 1982-2015 | SLA 1993/2015 | Taux 1979-2015 | Tauy 1979-2015 | P 1979-2015 | WD ST: 1960-2010 CCV satellites: 1996-2015 CCV in situ: 1960-2002 |
|---|---|---|---|---|---|---|
| El Niño events | Increase basin-wide during EP | Decrease mostly in the East | Increase north of 14°N, decrease to the South | Increase north of 6°N, decrease to the South | Decrease in the South-East region | CCV: Decrease of water discharge |
| Trends per decade | +0.24°C | +41 mm | -0.48 to +0.18 m/s | -0.5 to +0.1 m/s | -6.8 mm/d | ST: -520 m²/s CCV satellites: -1220 m³/s CCV in situ: -1340 m³/s |

Table 3: Summary of the main modifications of five ECVs analyzed, as observed in the SCS
during El Niño events and on basin-averaged long-term trends during the reported years.
Interestingly, it has been found that each ECV responds to specific ENSO types in
different ways. The first EOF mode on SST (75% of the variance) revealed that Eastern
Pacific El Niño types trigger a basin-wide warming of the SST with a 6-month lag. The second
EOF mode on SST indicated a clear correlation with all types of ENSO events (represented by
the SST Niño4 index). The first EOF mode on SLA (48% of the variance) showed that the SLA
falls throughout the basin during an El Niño event, with a 3-month lag, and more intensely in
the eastern part of SCS. Unexpectedly, the SST and SLA thus have an inverse response to
ENSO. This was examined by Rong et al. (2007). Using the subsurface temperature analysis
from Ishii et al. (2006), they show the coexistence of positive temperature anomalies in the
upper 75 m (hence >0 SST anomalies) and negative temperature anomalies below 75 m (at
least down to 700 m) during El Niño events. These out-of-phase temperature anomalies (0-

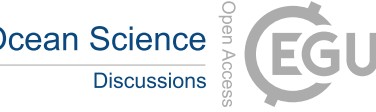

75 m versus 75-700 m) result in negative anomalies of the thermosteric sea level
component, which account for most of the negative SLA during El Niño events.

Previous studies have identified different potential hypotheses to explain the

anomalous ENSO-related SST changes. Wang et al. (2000) attributed the SCS warming to an
anomalous lower-tropospheric anti-cyclonic circulation located in the western North Pacific
through an 'atmospheric bridge'. The anomalies can persist until the following summer. Qu
et al. (2004) examined the heat transport through the Luzon Strait from the Pacific into the
SCS and suggested that this 'oceanic bridge' between the two basins was a key process in
conveying the impact of the Pacific ENSO into the SCS (both for SST and SLA). The respective
parts of the atmospheric and oceanic bridges clearly remain to be clarified.

The first EOF mode (64% of the total variance) of the zonal wind component revealed

that the wind increased eastwards north of 14°N (decreased south of 14°N) during an El
Niño event, with a 5-month lag. The first EOF mode (49% of the variance) of the meridional
wind component showed a strengthening of the northward winds north of 6°N (weakening
south of 6°N) during an El Niño event, with a 3-month lag. When superimposed on the strong
mean seasonal cycle, the El Niño events hence produce a slowdown of the winter NE and
summer SW monsoon winds. For precipitation, the first EOF mode (48% of the variance)
showed a decrease in rainfall over the entire region during El Niño phases. The linkage of
near-coastal in situ P stations and ENSO appeared to be stronger at lower latitudes in
Vietnam, for instance with P decreasing by up to 70% relative to the mean at the Da Nang
station during El Niño phases. Finally, water discharges taken from the Red River (at least at
the ST station) before the impoundment of the Hoa Binh Dam showed correlation to ENSO
but the post-dam impoundment period did not show a clear correlation to ENSO. The water
discharge from the Mekong River (at least at the CCV station) seems to respond to ENSO via
reduced water flow (34% of the mean for the 1997-98 El Niño event) with a 7 to 8-month
lag.



The patterns of the trends presented in the previous section provide useful elements
to determine the long-term variability of the ECVs. SST has risen by 0.24°C/decade basin-
wide (except along the coastlines of China and central Vietnam). The SLA increased with a
mean rising rate of 4.1 mm/yr with the highest rates in the central and the eastern parts of
the SCS basin, which is approximately 3 times faster than the global rate of sea level rise.
This corroborates the need to include data from the SCS (and semi-enclosed basins in
general) to improve the computation of global mean sea level trends (Dieng et al., 2015). The
zonal wind decreased over the period 1970-2015 by up to 0.48 m/s/decade in the eastern
and western parts of the region. The meridional wind strengthened southwards over the
entire region, except in the eastern part of the basin where the northward winds increased
by up to 0.18 m/s/decade. The satellite-derived precipitation decreased over the period
(1979-2015) by approximately 0.7 mm/d/yr over almost the entire region. In situ near-
coastal P stations in Vietnam showed increasing trends at lower latitudes, while the
northern stations showed slightly increasing to decreasing trends. All of the water
discharges that were observed showed small relative decreasing trends with higher rates
(compare to the mean) at the Red River ST station (1.5% of the mean), which is located a few
kilometers upstream of the city of Hanoi.
As a cautionary note, as discussed in England et al. (2014) and others, it is crucial to
consider the likely role of the PDO in the calculations of long-term trends. An inversion of the
sign of the PDO (e.g., in 2000) can induce a strengthening of the Pacific trade winds, leading
to the slowdown of the Pacific Ocean surface warming that has been observable since 2001
and is related to the changes in the regionally analyzed ECVs. Furthermore, as discussed in
the previous section and as addressed by Solomon and Newman (2012), it is essential to
consider the modulation of long-term trends that is induced by ENSO events (even when
estimated over 30 years), especially in the Indo-Pacific region.



Our results are based on a unique and unprecedented long-term data set we thought
to be relevant to each ECV. To confirm these results, it would however be fair to analyze
complementary data sets such as NCEP (Kalnay et al., 1996) and JRA-55 (Kobayashi et al.,
2015) for P, and TropFlux (Kumar et al., 2013) for P, SST and surface wind. Furthermore, the
growing literature that is available on the subject, and particularly the literature mentioned
throughout the text, has enabled us to identify mechanisms that likely account for the
observed variability at different time scales. Strictly quantifying the (common or not)
mechanisms responsible for the observed anomalies (including the ones of the 2015-16 El
Niño not discussed here) and trends highlighted in this paper by using outputs from
ocean/atmosphere dynamic models validated with our observations seems crucial to better
understand the seasonal, interannual and long-term variability of the region. This will be the
subject of another study.

**Acknowledgments:**
We benefited from the freely available datasets for SST (ftp://eclipse.ncdc.noaa.gov/pub/OI-
daily-v2/),     SLA     (http://www.aviso/altimetry.fr/en/data/products/sea-surface-height-
products/global/msla-mean-climatology), P (http://www.ecmwf.int/en/research/climate-
reanalysis/era-interim) and WD (http://ctoh.legos.obs-mip.fr). We warmly thank Vietnam's
Hydro-Meteorological Service for providing the inland precipitation measurements and the
Mekong River Commission for providing water discharge data from the gauge stations in
Cambodia. We are also very thankful for having had the opportunity to use the Mean
Dynamical Topography data provided by AVISO. Fruitful discussions with Gael Alory, Sylvain
Biancamaria, Frédéric Frappart, and Sylvain Ouillon, from LEGOS, help us to improve our
analysis.



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

Preliminary assessment of SARAL/AltiKa observations over the Granges-
Brahmaputra and Irrawaddy Rivers. Mar. Geod. 38(S1): 568-580, doi:
10.1080/01490419.2014.990591

Frappart F., Legrésy B., Niño F., Blare F., Fuller N., Fleury S., Birol F., Calmant S. (2016) An
ERS-2 altimetry reprocessing compatible with ENVISAT for long-term land and ice
sheets studies. Remote Sens. Environ. 184: 558-581, doi: 10.1016/j.rse.2016.07.037.
GCOS (2016). The Global Climate Observing System: implementation needs, GCOS-200,
GOOS-214. World Meteorological Organization, 325 pages (Available at:
http://unfccc.int/files/science/workstreams/systematic_observation/application/pd
f/gcos_ip_10oct2016.pdf)
Gobin A., Nguyen H. T., Pham V. Q., Pham H. T. T. (2016) Heavy rainfall patterns in Vietnam
and their relation with ENSO cycles. Int. J. Climatol. 36: 1686-1699, doi:
10.1002/joc.4451.
Hanley D. E., Bourassa M. A., O'Brien J. J., Smith S. R., Spade E. R. (2003) A quantitative
evaluation of ENSO indices. J. Climate, 16: 1249-1258, doi:
http://dx.doi.org/10.1175/1520-0442(2003)16<1249:AQEOIEI>2.O.CO
Hannachi A., Jolliffe I. T., Stephenson D. B. (2007) Empirical orthogonal functions and related
techniques in atmospheric sciences: A review. Int. J. Climatol., 27, 1119-1152, doi:
10.1002/joc.1499.
Ho C.-R., Zheng Q., Soong Y. S., Kuo N.-J., Hu J.-H. (2000) Seasonal variability of sea surface
height in the South China Sea observed with TOPEX/Poseidon altimeter data. J.
Geophys. Res. 105: 13981-13990, doi: 10.1029/2000JC900001.
Huynh H.-N., Alvera-Azcarate A., Barth A., Beckers J.-M. (2016), Reconstruction and analysis
of long-term satellite-derived sea surface temperature for the South China Sea. J.
Oceanogr., 1-20, doi: 10.1007/s10872-016-0365-1.



IPCC Fourth Assessment Report (AR4): Contribution of Working Group I, II and III to the
Fourth Assessment Report of the Intergovernmental Panel on Climate Change. Core
Writing Team, Pachauri R. K. And , Reisinger A. (Eds), IPCC, Geneva, Switzerland. Pp
104.

Ishii, M., Kimoto, M., Sakamoto, S., Iwasaki, S.-I. (2006) Steric sea level changes estimated
from historical ocean subsurface temperature and salinity analyses. J. Oveanogr., 62
(2), 155-170.

Juneng L., Tangang F. T. (2005), Evolution of ENSO-related rainfall anomalies in Southeast
Asia region and its relationship with atmosphere-ocean variations in Indo-Pacific
sector. Clim. Dynam., 25: 337-350, doi: 10.1007/s00382-005-0031-6.

Kalnay E, et al. (1996) The NCEP/NCAR 40-year reanalysis project. Bulletin of American
Meteorology Society, 77(3), 437-471, doi: https://doi.org/10.1175/1520-
0477(1996)077<0437:TNYRP>2.0.CO;2.

Klein S. A., Soden B. J., Lau N.-C. (1999), Remote sea surface temperature variations during
ENSO: Evidence for a tropical atmospheric bridge. J. Climate, 12, 917-932, doi:
http://dx.doi.org/10.1175/1520-0442(1999)012<0917:RSSTVD>2.0.CO;2.

Kobayashi S, et al. (2015) The JRA-55 Reanalysis: General specifications and basic
characteristics. Journal of the Meteorological Society of Japan, 93, 5-,48, doi:
http://doi.org/10.2151/jmsj.2015-001.

Kouraev A. V., Zakharova E. A., Samain O., Mognard N. M., Cazenave A. (2004), Ob's river
discharge from TOPEX/Poseidon satellite altimetry (1992-2002). Remote Sens.
Environ., 93, 238-245, doi: http://doi.dx.org/10.1016/j.rse.2004.07.007.

Kumar Praveen B., Vialard J., Lengaigne M., Murty V. S. N., McPhaden M.J. (2012) TropFlux:
air-sea fluxes for the global tropical oceans – description and evalution. Clim. Dyn. 38,
1521, doi:10.1007/s00382-011-1115-0.

Leuliette E. W., Willis J. K. (2011), Balancing the sea level budget. Oceanography. 24(2):122-
129., doi:10.5670/oceanog.2011.32

Liu Q., Jiang X., Xie S.-P., Liu W. T. (2004) A gap in the Indo-Pacific warm pool over the South
china Sea in borel winter: Seasonal development and interannual variability. J.
Geophys. Res., 109(C07012), doi:10.1029/2003JC002179.

Lu X.X, Siew R.Y. (2006), Water discharge and sediment flux changes over the past decades
in the Lower Mekong River: possible impact of the Chinese dams. Hydrol. Earth. Syst.
Sci., 10: 181-195, doi: hal-00304834.

Manton M. J., Della-Marta P. M., Haylock M. R., Hennessy K. J., Nicholls N., Chambers L. E.,
Collins D. A., Daw G., Finet A., Gunawan D., Inape K., Isobe H., Kestin T. S., Lefale P.,





Leyu C. H., Lwin T., Maitrepierre L., Ouprasitwong N., Page C. M., Pahalad J., Plummer
N., Salinger M. J., Suppiah R., Tran V. L., Trewin B., Tibig I., Yee D. (2001), Int J
Climatol., 21:269-284, doi: 10.1002/joc.610.
Mantua N. J., Hare S. R. (2002), The Pacific Decadal Oscillation. J. Oceanogr.. 58(1), 35-44.
Mei W., Xie S.-P. (2016), Intensification of the landfalling typhoons over the northwest
Pacific since the late 1970s. Nat. Geosci., 9: 753-758, doi:10.1038/NGEO2792.
Meyssignac B., Llovel W., Cazenave A., Salas-Melia D., Becker M. (2012) Tropical Pacific
spatial trend patterns in observed sea level: internal variability and/or anthropogenic
signature? Climate of the Past. 8:787-802. doi: 10.5194/cp-8-787-2012
Nguyen D.-Q., Renwick J., McGregor J. (2014), Variations of surface temperature and rainfall
in Vietnam from 1971 to 2010. Int. J Climatol., 34, 249-264, doi: 10.1002/joc.3684.
North G. R., Bell T. L., Cahalan R. F., Moeng F. J. (1982), Sampling Errors in the Estimation of
Empirical Orthogonal Functions. Amer. Meteor. Soc., 110: 699-706, doi:
http://dx.doi.org/10.1175/1520-0493(1982)110<0699:SEITEO>1.0.CO;2.
Peng D., Palanisamy H., Cazenave A., Meyssignac B. (2013), Interannual Sea Level Variations
in the South China Sea over 1950-2009. Mar. Geod., 36: 164-182, doi:
http://dx.doi.org/10.1080/01490419.2013.771595.
Qu T. (2001) Role of the ocean dynamics in determining the mean seasonal cycle of the
South China Sea surface temperature. J Geophys. Res., 106: 6943-6955, doi:
10.1029/2000JC000479.
Qu T., Kim Y. Y., Yaremchuk M., Tozuka T., Ishida A., Yamagata T. (2004) Can Luzon Strait
Transport play a role in conveying the impact of ENSO to the South China Sea? J.
Climate, 17: 3644-3657, doi: https://doi.org/10.1175/1520-
0442(2004)017<3644:CLSTPA>2.0.CO;2.
Qu T., Song T., Yamagata T. (2009) An introduction to the South China Sea throughflow : its
dynamics, variability and application for climate. Dyn. Atmosph. Oceans, 47, 3-,14.
Räsänen T. A., Kummu M. (2013), Spatiotemporal influences of ENSO on precipitation and
flood pulse in the Mekong River Basin. J of Hydrol., 476, 154-168, doi:
http://dx.doi.org/10.1016/j.hydrol.2012.10.028.
Räsänen T. A., Lindgren V., Guillaume J. H. A., Buckley B. M., Kummu M. (2016) On the spatial
and temporal variability of ENSO precipitation and drought teleconnection in mailand
Southeast Asia. Clim. Past., 12, 1889-1905, doi: 10.5194/cp-12-1889-2016..
Rayner N. A., Parker D. E., Horton E. B., Folland C. K., Alexander L. V., Rowell D. P., Kent E. C.,
Kaplan A. (2003), Global analysis of sea surface temperature, sea ice, and night




marine air temperature since the late nineteeth century. J. Geophys. Res., 108(D14):

4407, doi: 10.1029/2002JD002670.

Reynolds R. W., Smith T. M., Liu C., Chelton D. B., Casey K. S., Schlax M. G. (2007) Daily High-

Resolution-Blended Analyses for Sea Surface Temperature. American Meteorological

Society. 20: 5473, 5496, doi: http://dx.doi.org/10.1175/2007/JCLI1824.1.

Rio M.-H., Mulet S., Picot N. (2014), Beyond GOCE for the ocean circulation estimate:

Synergetic use of altimetry, gravimetry, and in situ data provides new insight into

geostrophic and Ekman currents. Geophys. Res. Lett., 41: 8918-8925, doi:

10.1002/2014GL061773.

Rong Z., Liu Y., Zong H., Cheng Y. (2007), Interannual sea level variability in the South China

Sea and its response to ENSO. Global Planet. Change, 55, 257-272, doi:

http://dx.doi.org/10.1016/j.gloplacha.2006.08.001.

Saji N. H., Goswami B. N., Vinayachandran P. N., Yamagata T. (1999),  A,  dipole mode in the

tropical Indian Ocean. Nature, 401: 360-363.

Sing A., Delcroix T., Cravatte A. (2011) Contrasting the flavors of El Niño Southern Oscillation

using sea surface salinity observations. J. Geophys. Res., 116: C06016,

doi:10.1029/2010JC006862.

Solomon A., Newman M. (2012), Reconciling disparate twentieth-century Indo-Pacific ocean

temperature trends in the instrumental record. Nat. Clim. Change., 2: 691-697, doi:

10.1038/nclimate1591.

Tan W., Wang X., Wang W., Wang C., Zuo J. (2016) Different Responses of Sea Surface

Temperature in the South China Sea to Various El Niño Events during Boreal Autumn.

934        J. Climate., 29, 1127-1142, doi: http://dx.doi.org/10.1175/JCLI-D-15-0338.1.

Trenberth K. E., Hoar T. J. (1997) El Niño and climate change. Geophys. Res. Lett. 24: 3057-

3060, doi: 10.1020/97GL03092.

Trenberth K. E., Stepaniak D. P. (2001) Indices of El Niño Evolution. J. Climate., 14 : 1697-

1701., doi: http://dx.doi.org/10.1175/1520-

0442(2001)014<1697:LIOENO>2.0.CO;2.

Tuen K. L. (1994) Monitoring of sea surface temperature in the South China Sea.

Hydrobiologia. 285: 1-5, doi: 10.1007/978-94-011-0958-1_1.

Vinh V. D., Ouillon S., Thanh T. D., Chu L. V. (2014), Impact of the Hoa Binh dam (Vietnam) on

water and sediment budgets in the Red River basin and delta. Hydrolg. Earth Sci., 18:

3987-4005, doi: 10.5194/hess-18-3987-2014.



Wang B., LinHo (2002), Rainy season of the Asian-Pacific summer monsoon. J. Climate., 15,
386-398, doi:
http://dx.doi.org/10.1175/15200442(2002)015<0383:RSOTAP>2.0.CO;2.
Wang B.,  H, uang F., Wu Z., Yang J., Fu X., Kikuchi K. (2009)  Multi-scale climate variability of
the South China Sea monsoon: A review. Dyn. Atmosph. Oceans, 47, 3-,14
Wang B., Wu R., Fu X. (2000) Pacific-East Asian Teleconnection: How does ENSO affect East
Asian climate? J. Climate, 13: 1517-1536, doi: https://doi.org/10.1175/1520-
0442(2000)013<1517:PEATHD>2.0.CO;2
Wang C., Wang W., Wang D., Wang Q. (2006), Interannual variability of the South China Sea
associated with El Niño. J. Geophys. Res., 111: C03023, doi: 10.1029/2005JC003333.
Wang C., Xie S.-P., Carton J. A. (2004) A global survey of ocean-atmosphere interaction and
climate variability, in *Earth Climate: The Ocean-Atmosphere Interaction* edited by
Wang C et al., 1-19, AGU, Washington D C.
Wyrtki, K., 1961: Physical oceanography of the Southeast Asian waters. Naga Rep. 2, 195 pp.
[Available from: https://escholarship.org/uc/item/49n9x3t4].
Wu C.-H., June Chang C.-W. (2005) Interannual variability of the South China Sea in a data
assimilation model. J. Geophys. Res., 32:LI7611, doi:10.1029/2005GL023798
Wu L., Wang B., Geng S. (2005) Growing typhoon influence on east Asia. Geophys. Res. Lett.,
32 (18), LI8703, doi:10.1029/2005GL022937.
Yang Y., XIe S.-P., Du Y., Tokinaga H. (2015) Interdecadal difference of interannual variability
characteristics of South China Sea SSTs associated with ENSO. Amer Mereor Soc. 28:
7145-7160, doi:10.1175/JCLI-D-15-0057.1
Xue Z., Liu J. P., Ge Q. (2011) Changes in hydrology and sediment delivery of the Mekong
River in the last 50 years: connection to damming, monsoon, and ENSO. Earth Surf.
Proc. Land., 36: 296-308, doi: 10.1002/esp.2036.
Zhuang W., Xie S.-P., Wang D., Taguchi B., Aiki H., Sasaki H. (2010) Intraseasonal variability in
sea surface height over the South China Sea. J. Geophys. Res., 115: C04010,
doi:10.1029/2009JC005647.