# Peer review of "Seasonal and interannual (ENSO) climate variabilities and trends in the South China Sea over the last three decades"

_Ocean Science, 2017_

## Referee Comment (RC1) · Anonymous Referee #1 · 14 Feb 2018

General Remarks:

The paper discusses the development of five so called "Essential Climate Variables" which have been observed for the South China Sea over the last three decades. Four aspects: i) mean and standard deviation ii) seasonal variability iii) inter-annual variability iv) trends and five parameters: i) SST ii) SLA iii) precipitation iv) surface wind v) water discharge are considered for this study. All of these data are based on observational data, if one accepts the point that an optimal interpolation or a re-analysis is also closely related to observations. Although the paper definitely contains a lot of interesting material which would deserve its publication, unfortunately, in the current

state of the manuscript it must be rejected.

The main criticism is that the paper is written like a report and not like a scientific research paper. The authors follow a very rigid structure, where for each of the four aspects each of the five parameters is discussed separately, providing insufficient or only scarce overarching information. Due to this rigid scheme, a lot of unnecessary or even redundant information is provided. For this reason the paper in relation to its content is much too long and boring to read with in total 48 manuscript pages. A good example of the problems which arise from the report-like structure is the fact, that annual mean SST and wind speeds are presented despite the circumstance that in such a monsoon dominated area these quantities are more or less meaningless, since in one year the oceanic and atmospheric system switches between two dominant modes. In this case a mean for the summer and winter monsoon situation would be much more valuable. The annual mean patterns just are just synthetic distributions which have no representation in the real world.

Another very striking problem which arises from this rigid structure can be seen, when looking at the standard deviation of the SSTs or the u- and v-winds (Figs. 2b, 3b, 3d). These figures are nearly similar to those of their first EOFs (Figs. 4a, 6a, 6c), which certainly has to be expected. Interestingly, this close agreement between these figures has not even been mentioned by the authors. This is a good example how the authors just treat each aspect separately, without considering any of the cross-connections, which in many cases would provide a lot of additional scientific information. Another general criticism concerns the way how the authors interpret the results of their analyses. Most of these explanations are just speculations or even platitudes, which are rarely being substantiated by adequate information and/or literature. Here are just two examples:

Lines 293-301: the development of the second SST mode is explained basically by the monsoonal variations, although it only explains 5% of the total variance. Moreover, solar radiation and ocean heat transport are both brought into play, without giving any

indication of their particular contribution. One would expect that also the first EOF is governed by monsoonal variations. It seems that the monsoon triggers two orthogonal modes at the same time. However, this interesting question has not even been mentioned by the authors.

Lines 377-381: It is obvious that the river discharge depends on the monsoon dynamics. Therefore the peak run-off should occur during the rainy season, which is no surprise. However, a time lag may occur, since firstly, it will take a few weeks from some of the catchment areas to the Mekong River mouth, and moreover, the onset and peak time of the rainy season will vary for the different parts of the large Mekong catchment area.

I have noted a number of further minor points of criticism regarding some specific details of the manuscript. However, since according to my opinion the entire structure of the paper has to be changed, at present it makes no sense to list them here.

In conclusion, I would propose that the authors rewrite the paper using the style of a research paper. This means that they construct the paper around their major finding. For this reason only information supporting these findings should be presented in a stringent way. Otherwise the reader just becomes confused or bored by too many unnecessary details.

The other big drawback of the paper is, that due to its report-like structure it was not possible to answer the most obvious and pressing questions for this research area, i.e., whether the summer or winter monsoon have increased or decreased in strength over the last decades, and if the onset time of both monsoon phases has changed, as speculated by many authors. I can imagine that the monthly resolution of the data could be problematic to answer the latter questions. However, since this question is of vast interest for the entire region, the authors should at least make an attempt to answer this question.

---

## Referee Comment (RC2) · Anonymous Referee #2 · 27 Feb 2018

Comments on "Seasonal and interannual (ENSO) climate variabilities and trends in the Southern China Sea over the last three decades"

Authors: Violaine Piton and Thierry Delcroix

General comments:

The manuscript by Piton and Delcroix analyze the variability of 5 parameters (sea surface temperature, sea level anomaly, surface wind components, precipitation and water discharge) over the South China Sea using relatively long datasets. They have found the impact of ENSO variations to the analyzed variables with the corresponding time-lag between the impact.

[Figure]

In general the quality of the paper is good – the description of data and methods are followed by the analysis of the parameters on the seasonal scale (Section 4) and then on interannual scale (Section 5). The trends of the variables are presented in Section 6 and afterwards Conclusion and discussion is presented in Section 7.

I think the authors have done great work collecting and presenting (discussing) results from other's work relevant to this study, but in order to be "true" review paper for the area, emphasize should be on other's results instead of the (new) findings that are presented in this manuscript. I have put parentheses around new as if there are already so many papers from different authors about the trends and variability of essential climate parameters in the area, what is new in this paper? Coherent methodology or newer dataset should not be considered worthy for new knowledge – perhaps it is also important to emphasize new findings in the text. Although the paper was interesting to read, it needs some revision before it can be accepted for publication in Ocean Science. I am not very satisfied how the Conclusion and Discussion is presented – the authors repeat most of the already presented results and discussion (from Sections 3-6) in a shorter way. I would suggest renaming the section to "Conclusions" and bring out important information or conclusions in this section. Considering that the manuscript is already written in a way, where the discussion is embedded in the results section, I think it is reasonable to skip the discussion from the conclusions.

Some comments:

Lines 5-6: Please rephrase "..., and P increases in the north". In the beginning of the sentence the authors discuss the changes of SST and SLAs that occur in the summer and then end the sentence about changes that occur in geographical space.

Line 17: "... Pacific Decadal Oscillation (PDO) ..." I missed the description of PDO in later manuscript.

Fig 1: Please add coastline preferably with thick black line.

Line 51: "... as well as by the water exchange with the surrounding ..."

Table 1: I would suggest another plot about the location of the stations colored either by the mean or standard deviation and other important information shown as a text close to the station.

Line 151: "... were obtained ..."

Line 163: I assume correlation or determination coefficient was meant instead of slope of the regression.

Line 192: "... EOF modes greater equal to two ..."

Line 201: " ... are available both over the ocean and land."

Line 206: Please rephrase "... in the winter in the area."

Fig 3: I would suggest adding mean and std of wind-speed velocity as sqrt(u**2+v**2)

Fig 4: Why to write in the caption: "The product between spatial and temporal functions denote anomalous SST (in $^oC$) and SLA (in $m$) respective to the mean values." These products are never shown in the paper and as expected, the largest EOF mode describing the largest part of the variance, should show variance. I assume anomalous SST and SLA respective to the mean values is the variance.

Lines 309-314: I miss the importance of this section. Although the authors refer to Qu (2001) and the methodology using mixed layer depths, they present the time-series of SST. I do not expect mixed layer depths to be positively correlated with SSTs.

Line 327: "... correspond to the maximum... "

Line 354: Where can I see the value 6 mm/d ? I assume it is seen from the Fig. 3f, please indicate this in the text.

Table 2: Where did you get the values for Niño1+2, Niño3.4, Niño4, SOI and EMI? I saw references to different authors on page 9 , did you get the values from those

papers?

Line 409: Where can I see increasing values 0.7 and 1.0 $^oC$?

Line 425 and 426: Where can I see values 0.1 and 0.2 $^oC$?

Line 451: I guess values 5-10 cm are seen on Fig. 2d, please indicate in the text.

Fig. 13: I am not sure about the panel (e) as the authors only mention it on lines 619-621. The authors do not discuss the vector representation – are there some changes in the directions of the winds as well? What can we learn from the vector representation of the trends?

Line 601: PDO comes in with no previous indication what this is. I assume it is Pacific Decadal Oscillation from the Abstract.

Line 629: Satellite? I thought the precipitation was from ERA interim re-analysis, which is model product.

Section 7 is definitely too long and repeats already shown results in not a good matter. I suggest rewriting it in a short and concise way, where the authors bring out the most important results or conclusions of their work.

Please also note the supplement to this comment:
https://www.ocean-sci-discuss.net/os-2017-104/os-2017-104-RC2-supplement.pdf

---

## Referee Comment (RC3) · Anonymous Referee #3 · 3 Mar 2018

The paper discusses the so called "Essential Climate Variables" in the area of the South China Sea over the last three decades. Included are mean and standard deviation, seasonal and inter-annual variability, and trends for five ECVs. The ECVs included are: SST, SLA, precipitation, surface wind and water discharge.

Although the paper definitely contains some interesting aspects, which might be worth a publication I cannot recommend publication in its current version for several reasons.

My main criticism is that the authors seem to be not convinced of their own results and somewhat repeat and hide behind the results/finding of others, which is mirrored in the length of the paper and many redundancies. I do think also it is because of the

repetitions too long and it contains too many figures, which are partly meaningless, because nothing new is shown. This is strikingly clear in case of the e.g. reversing signals in SST and wind on annual scale. In a monsoon driven system not really surprising, worth presenting, no, because nothing new. This is just one example.

I miss also connection (in presentation and/or discussion) between the parameters studied, since they are not disconnected from each, but instead the reader get presented much information which is known, established and therefore the detailed presentation of their results e.g for the SST and wind pattern, is again redundant. Both the annual mean SST and wind speeds are presented despite the circumstance that in a monsoon dominated area these quantities are more or less meaningless, since in one year both the oceanic and atmospheric system alternates between two dominant phases.

My another main criticism is regarding precipitation and river discharge, as the authors just include Vietnamese data. Why contributions from China, which limits SCS to a huge extent, are not included at all? Not a single word is mentioning this, and I do think Pearl River might have a significant river discharge into SCS?

I have noted a number of additional minor points, but since I'm not recommending publication at this stage it is not worth listing them now, as the authors need to rewrite the entire manuscript in a short, precise way omitting all redundancies, but highlighting their results instead.

I'm sorry I cannot be more positive at this moment.

---

## Author Comment (AC1) · 29 Mar 2018

We do thank Referee#1 for his/her careful reading of our manuscript and relevant comments. Below are his/her comments (in italics), followed by our responses and description of related changes in manuscript.

*General Remarks:*

*The paper discusses the development of five so called "Essential Climate Variables" which have been observed for the South China Sea over the last three decades. Four aspects: i) mean and standard deviation ii) seasonal variability iii) inter-annual variability iv) trends and five parameters: i) SST ii) SLA iii) precipitation iv) surface wind v) water discharge are considered for this study. All of these data are based on observational data, if one accepts the point that an optimal interpolation or a re-analysis is also closely related to observations. Although the paper definitely contains a lot of interesting material which would deserve its publication, unfortunately, in the current state of the manuscript it must be rejected.*

*The main criticism is that the paper is written like a report and not like a scientific research paper. The authors follow a very rigid structure, where for each of the four aspects each of the five parameters is discussed separately, providing insufficient or only scarce overarching information. Due to this rigid scheme, a lot of unnecessary or even redundant information is provided. For this reason the paper in relation to its content is much too long and boring to read with in total 48 manuscript pages. A good example of the problems which arise from the report-like structure is the fact, that annual mean SST and wind speeds are presented despite the circumstance that in such a monsoon dominated area these quantities are more or less meaningless, since in one year the oceanic and atmospheric system switches between two dominant modes. In this case a mean for the summer and winter monsoon situation would be much more valuable. The annual mean patterns are just synthetic distributions which have no representation in the real world.*

In re-reading the submitted manuscript (ms.) we agree that it was too long, with some unnecessary and/or redundant information. We also agree that parts of the ms. may look like a report, although it is not that clear to us where the frontier between a report and a scientific research paper is. Whatsoever it was, a report-like or scientific paper, we appreciate that it generated a lot of interest with more than 400 views and downloads since early January.

In agreement with referee#1, we noted in the submitted ms., lines 343-345, that "The interpretation of the annual mean surface wind in a region that is highly influenced by strong seasonal wind reversals due to the monsoon cycles (see section 4) does not mean very much". To better consider the issue in the revised ms., we have removed all paragraphs dealing with the mean patterns (and standard deviations) of the surface winds, SST and precipitation, and rather focused on the two most contrasted seasons, JFM and JJA. We chose to keep the description of the altimeter-derived mean dynamic topography and mean water discharge (and related standard deviations) because, to our knowledge, it was never documented in the literature.

In doing this, we have deleted Figures 2a-b, 3a-f, 4a-f, 5, and 6a-f, and add the new Figure 4a-f showing the mean JFM and JJA surface wind (in vector form), SST and Precipitation. That new figure is shown below. We also have combined the former Section 3 'ECV means and standard deviations' and former Section 4 'Seasonal variability' into a unique Section 3 mostly providing a

quick description of the two JFM and JJA seasons as a background. The combination of the former Sections 3 and 4 into a unique and more concise Section 3 results in a reduction of 6 pages.

*Another very striking problem which arises from this rigid structure can be seen, when looking at the standard deviation of the SSTs or the u- and v-winds (Figs. 2b, 3b, 3d). These figures are nearly similar to those of their first EOFs (Figs. 4a, 6a, 6c), which certainly has to be expected. Interestingly, this close agreement between these figures has not even been mentioned by the authors. This is a good example how the authors just treat each aspect separately, without considering any of the cross-connections, which in many cases would provide a lot of additional scientific information.*

As noted in the submitted ms., lines 85-86, the standard deviation maps were presented because they denote the "overall variability", i.e., mostly the seasonal and interannual changes in our study. The similarity between the standard deviations and the seasonal EOF of the SSTs, u- and v-winds is indeed clear from the presented figures, although these have to be shown first. We agree we should have stressed that similarity, indicating that the overall variability mostly results from the seasonal variability. The point no longer arises for the surface wind, SST and P since we have removed the standard deviation and seasonal EOF figures. The similarity is mentioned for the SLA in the revised ms.

*Another general criticism concerns the way how the authors interpret the results of their analyses. Most of these explanations are just speculations or even platitudes, which are rarely being substantiated by adequate information and/or literature. Here are just two examples*:

*Lines 293-301: the development of the second SST mode is explained basically by the monsoonal variations, although it only explains 5% of the total variance. Moreover, solar radiation and ocean heat transport are both brought into play, without giving any indication of their particular contribution. One would expect that also the first EOF is governed by monsoonal variations. It seems that the monsoon triggers two orthogonal modes at the same time. However, this interesting question has not even been mentioned by the authors.*

As we have drastically reduced the analysis of seasonal changes, removing EOF analysis and concentrating on the JFM and JJA seasons only, discussion of the second seasonal EOF mode on SST no longer appear in the revised ms.

*Lines 377-381: It is obvious that the river discharge depends on the monsoon dynamics. Therefore the peak run-off should occur during the rainy season, which is no surprise. However, a time lag may occur, since firstly, it will take a few weeks from some of the catchment areas to the Mekong River mouth, and moreover, the onset and peak time of the rainy season will vary for the different parts of the large Mekong catchment area.*

We agree with that comment. Note, however, that the former Figure 9 (now Figure 3) not only provide information on the timing of the seasonal water discharge but also on the amplitude. Moreover, as was discussed in the last paragraph of Interannual Variability section, it is likely that the regulation of water flow by the barrages further impacts the time lag between P in the catchment areas and the river discharge. A comprehensive study of the relationships between the monsoon dynamics and the seasonal river discharges is beyond the scope of our ms.

*I have noted a number of further minor points of criticism regarding some specific details of the manuscript. However, since according to my opinion the entire structure of the paper has to be changed, at present it makes no sense to list them here.*

*In conclusion, I would propose that the authors rewrite the paper using the style of a research paper. This means that they construct the paper around their major finding. For this reason only information supporting these findings should be presented in a stringent way. Otherwise the reader just becomes confused or bored by too many unnecessary details.*

Following the reviewer's comment, we drastically reduce the analysis of mean values, standard deviations, and seasonal variabilities of the five analysed ECV, concentrating now on interannual (ENSO) variability and long-term trends.

*The other big drawback of the paper is, that due to its report-like structure it was not possible to answer the most obvious and pressing questions for this research area, i.e., whether the summer or winter monsoon have increased or decreased in strength over the last decades, and if the onset time of both monsoon phases has changed, as speculated by many authors. I can imagine that the monthly resolution of the data could be problematic to answer the latter questions. However, since this question is of vast interest for the entire region, the authors should at least make an attempt to answer this question.*

We agree with the reviewer that we are not in a position to solve the question of the onset time of the monsoon with the monthly data sets we have in hand. To the best of our knowledge, this is a long-standing question, and we had no pretention to even partly solve it in our ms. We however better linked our results to the main 'usual' monsoon features when addressing ENSO and long-term trends in the Abstract, in Sections 4 and 5, and in the Summary and Conclusion section with a better description of the former Figure 8 and the new Figure 10. For instance, in the revised abstract, we note in line with:

i) the former Figure 8 (shown below): '….. The winter N-NE and summer S-SW monsoon winds weaken during El Niño events. Opposite wind anomalies are observed during La Niña events, enhancing both winter and summer monsoons….',

ii) the new Figure 10 (shown below): ' ….. Increasing trends in northerly winds are observed in both winter and summer seasons, potentially intensifying/weakening the winter/summer monsoon, respectively…'.

[Figure]

Figure 4. Mean surface wind (top panels), SST (middle panels, units are °C) and P (bottom panels, units are mm/day) in January-February-March (left panels) and June-July-August (right panels). The arrows in a-b denote the wind vectors, and the longest arrows equal 15 m/s. The black markers on panel (e, f) denote, from north to south, the location of the 17 inland rainfall stations and the crosses denote the location of Lục Yên, Đà Nẵng and Mỹ Tho discussed in the text (see Table 1). The red dots denote the location, from north to south, of the gauge stations at the Red River ST and Mekong CCV stations. The color codes differ between the figures.

[Figure]

Figure 8. El Niño composites of the anomalous surface wind vectors in JFM (a) and JJA (b) computed for 1983, 1987, 1992, and 1998, representing the 1982-83, 1986-87, 1991-92, and 1997-98 El Niño events, respectively. The arrows denote the vectors, with the longest arrows equal to 1 m/s.

[Figure]

Figure 10. Surface wind trends in (a) JFM and (b) JJA. The longest arrow is equal to 1.5x10-2 m/s/decade.

---

## Author Comment (AC2) · 29 Mar 2018

We do thank Referee#2 for his/her careful reading of our manuscript and relevant comments. Below are his/her comments (in italics), followed by our responses and description of related changes in manuscript.

*General comments:*

*The manuscript by Piton and Delcroix analyze the variability of 5 parameters (sea surface temperature, sea level anomaly, surface wind components, precipitation and water discharge) over the South China Sea using relatively long datasets. They have found the impact of ENSO variations to the analyzed variables with the corresponding timelag between the impact.*

*In general the quality of the paper is good – the description of data and methods are followed by the analysis of the parameters on the seasonal scale (Section 4) and then on interannual scale (Section 5). The trends of the variables are presented in Section 6 and afterwards Conclusion and discussion is presented in Section 7.*

We thank the reviewer for his/her nice and encouraging words.

*I think the authors have done great work collecting and presenting (discussing) results from other's work relevant to this study, but in order to be "true" review paper for the area, emphasize should be on other's results instead of the (new) findings that are presented in this manuscript. I have put parentheses around new as if there are already so many papers from different authors about the trends and variability of essential climate parameters in the area, what is new in this paper? Coherent methodology or newer dataset should not be considered worthy for new knowledge – perhaps it is also important to emphasize new findings in the text. Although the paper was interesting to read, it needs some revision before it can be accepted for publication in Ocean Science.*

We removed the term "short review" in the Abstract and replace it by "integrated analysis" which we believe is more relevant to what our ms. really is. We still do believe our integrated analysis is very valuable despite the fact – and even due to the fact – there are already many papers dealing with similar themes. As a matter of fact, as we noted for Reviewer#1, we appreciate that our ms. generated a lot of interest with more than 400 views and downloads since early January despite the fact "there are already some many papers". Moreover, we all know that conclusion of scientific papers may sometime be dependent on used methodology and/or time series duration. Our conviction is then that the coherent methodology and enhanced time series duration we used are clearly appropriate and beneficial to a better understanding of the regional climate variability. We admittedly did not submit a fully original ms., but rather provide a concise co-analysis of five key ECVs (never done to the best of our knowledge) with recognition of already published results, if any.

*I am not very satisfied how the Conclusion and Discussion is presented – the authors repeat most of the already presented results and discussion (from Sections 3-6) in a shorter way. I would suggest renaming the section to "Conclusions" and bring out important information or conclusions in this section. Considering that the manuscript is already written in a way, where the discussion is embedded in the results section, I think it is reasonable to skip the discussion from the conclusions.*

We agree with that comment, and renamed the 'Conclusion and Discussion' section into 'Summary and Conclusion' section. The discussion of some of the results that appeared in previous sections has been moved to the Summary and Conclusion section.

*Some comments:*

*Lines 5-6: Please rephrase "..., and P increases in the north". In the beginning of the sentence the authors discuss the changes of SST and SLAs that occur in the summer and then end the sentence about changes that occur in geographical space.*

Results about the seasonal variability are much reduced in the revised ms. (we removed the seasonal EOF analysis and focus only on the JFM and JJA seasons, see our reply to referee #1), and hence no more appear in the Abstract (where Lines 5-6 were).

*Line 17: ": : : Pacific Decadal Oscillation (PDO) : : :" I missed the description of PDO in later manuscript.*

"....linked to the phase of the PDO...." was replaced by "linked to the possible influence of ENSO phase in the computation of long-term trends"

*Fig 1: Please add coastline preferably with thick black line.*

It is a question of appearance. We tested to drawn thick black lines for the coastline and found out the figure becomes less clear. Sorry, no change has been made here.

*Line 51: ": : : as well as by the water exchange with the surrounding ..."*

Changed.

*Table 1: I would suggest another plot about the location of the stations colored either by the mean or standard deviation and other important information shown as a text close to the station.*

The locations of the 17 selected inland rainfall stations are now plotted on Figures 4e-f for clarity. The modified Figure appears in our reply to referee #1. Note that the 17 selected stations are represented with black markers (the crosses represent the 3 stations discussed in the text, the 15 remaining are represented by the dots), the water discharges stations remain represented by the red dots.

*Line 151: ": : : were obtained ..."*

Corrected.

*Line 163: I assume correlation or determination coefficient was meant instead of slope of the regression.*

We confirm we meant 'the slope of the regression line' which represents the rate of change in y (satellite data) as x (in situ data) changes.

*Line 192: ": : : EOF modes greater equal to two ..."*

We meant equal to – or greater than – two.

*Line 201: " : : : are available both over the ocean and land."*

This line disappears, as we no more focus on mean and standard deviation.

Line 206: Please rephrase ": : : in the winter in the area."

This line disappears, as we no more focus on mean and standard deviation.

*Fig 3: I would suggest adding mean and std of wind-speed velocity as sqrt(u\*\*2+v\*\*2)*

We agree this could be useful. We however removed all paragraphs dealing with mean and standard deviation of the winds, following all reviewers' comments who noted that mean structures are meaningless in the SCS given the strong seasonal monsoon reversal.

*Fig 4: Why to write in the caption: "The product between spatial and temporal functions denote anomalous SST (in ₒC) and SLA (in m) respective to the mean values." These products are never shown in the paper and as expected, the largest EOF mode describing the largest part of the variance, should show variance. I assume anomalous SST and SLA respective to the mean values is the variance.*

That Figure 4 is removed in the revised ms. as we now focus on two contrasted seasons only (JFM and JJA). The sentence in the caption is however maintained in other figures when addressing the interannual EOF modes. Given the way we compute the EOF, we believe the sentence is necessary to indicate the units of the anomalous fields.

*Lines 309-314: I miss the importance of this section. Although the authors refer to Qu (2001) and the methodology using mixed layer depths, they present the time-series of SST. I do not expect mixed layer depths to be positively correlated with SSTs.*

These lines and the corresponding Figure 5 are removed in the revised ms.

*Line 327: ": : : correspond to the maximum: : : "*

We meant '…. the timing …. corresponds …'

*Line 354: Where can I see the value 6 mm/d ? I assume it is seen from the Fig. 3f, please indicate this in the text.*

This value could actually be view in computing 'the product between the spatial (Fig. 6e) and temporal (Fig. 6f) functions' in the seasonal EOF analysis. Line 354 no longer appears with the removal of the seasonal EOF analysis in the revised ms.

*Table 2: Where did you get the values for Niño1+2, Niño3.4, Niño4, SOI and EMI? I saw references to different authors on page 9 , did you get the values from those papers?*

Yes, these ENSO indices were referenced on page 9 in the paragraph called "climate indices" and the indices were extracted from these sources.

*Line 409: Where can I see increasing values 0.7 and 1.0 $_o$C?*

The maximum value in Figure 10a was about 0.06, and the EOF time function is about 8 in 1986-87 and 12 in 1997-98. Then the product between the space and time functions, quantifying the anomalies, is 0.48 (and not 0.7) and 0.72 (and not 1°C). This is corrected (we noted 0.5 and 0.7°C).

*Line 425 and 426: Where can I see values 0.1 and 0.2 $_o$C?*

The maximum value in Figure 10c was about +0.05 in the eastern half and -0.1 along the coast and south of Vietnam. The EOF time function in Figure 10d ranges within (-2, +2). Then the product between the space and time functions, quantifying the anomalies, is about 0.1 (0.05 x 2) in the east and 0.2 (0.1 x 2) along the coast.

*Line 451: I guess values 5-10 cm are seen on Fig. 2d, please indicate in the text.*

This is the same issue as above, and the reason why we noted in the caption that anomalies could be quantified computing the product between the EOF time and space functions. Looking at Figure 10e, maximum changes of the order of -0.025 occur in the east at about 15°N. Looking at Figure 10f, the EOF time function ranges within about -2 and +2. Hence the product between the space and time function reach about 5 cm (0.025 x 2 m). We changed 5-10 cm to 5 cm.

*Fig. 13: I am not sure about the panel (e) as the authors only mention it on lines 619-621. The authors do not discuss the vector representation – are there some changes in the directions of the winds as well? What can we learn from the vector representation of the trends?*

This vector representation was designed to ease the interpretation of trends in the figures showing trends in zonal and meridional components. We think this representation provide us with information about possible changes of the monsoon winds over time. From Figure 13e, and looking at Figure 7, there is in fact a hint for an intensification of the winter monsoon in the central and southern parts of the SCS.

To further our analysis, we plotted the trends for JFM and JJA over the period 1979-2015 (see the new Figures 10ab in our reply to Referee #1). As we note now in the text: "It appears that in winter, when comparing to Figure 7, there is an intensification of the northern winds along the coast of Vietnam, in the northernmost part of the SCS and in the Gulf of Thailand. These results seem to confirm the suspected intensification of the winter monsoon over the area. On the other hand, the linear trend of winds in JJA show an increasing tendency to winds blowing in opposite direction to monsoon winds, leading to a decreasing trend of the southern winds in summertime (when comparing to Figure 7), especially in the southern part of the SCS. These results are suggesting a decreasing in strength of the summer monsoon over the period considered."

*Line 601: PDO comes in with no previous indication what this is. I assume it is Pacific Decadal Oscillation from the Abstract.*

Thanks for noting this. The acronym is detailed in the revised version, and a reference is given.

*Line 629: Satellite? I thought the precipitation was from ERA interim re-analysis, which is model product.*

Yes, this is from a re-analysis. Corrected.

*Section 7 is definitely too long and repeats already shown results in not a good matter. I suggest rewriting it in a short and concise way, where the authors bring out the most important results or conclusions of their work.*

We removed the whole paragraph dealing with the seasonal cycle, and removed some sentences and related references in the last paragraph.

---

## Author Comment (AC3) · 29 Mar 2018

We do thank Referee#3 for his/her careful reading of our manuscript and relevant comments. Below are his/her comments (in italics), followed by our responses and description of related changes in manuscript.

*The paper discusses the so called "Essential Climate Variables" in the area of the South China Sea over the last three decades. Included are mean and standard deviation, seasonal and inter-annual variability, and trends for five ECVs. The ECVs included are: SST, SLA, precipitation, surface wind and water discharge. Although the paper definitely contains some interesting aspects, which might be worth a publication I cannot recommend publication in its current version for several reasons.*

*My main criticism is that the authors seem to be not convinced of their own results and somewhat repeat and hide behind the results/finding of others, which is mirrored in the length of the paper and many redundancies. I do think also it is because of the repetitions too long and it contains too many figures, which are partly meaningless, because nothing new is shown. This is strikingly clear in case of the e.g. reversing signals in SST and wind on annual scale. In a monsoon driven system not really surprising, worth presenting, no, because nothing new. This is just one example. I miss also connection (in presentation and/or discussion) between the parameters studied, since they are not disconnected from each, but instead the reader get presented much information which is known, established and therefore the detailed presentation of their results e.g for the SST and wind pattern, is again redundant. Both the annual mean SST and wind speeds are presented despite the circumstance that in a monsoon dominated area these quantities are more or less meaningless, since in one year both the oceanic and atmospheric system alternates between two dominant phases.*

We confirm we are convinced by our own results. We agree with the reviewer the ms. was too long. To shorten it, as responded to the other reviewers, we drastically reduce the number of pages, especially the ones appearing in the two former sections dealing with mean and standard deviations and seasonal EOF analysis. A brief description of the two JFM and JJA contrasted seasons is now given as a background in the revised ms.

We agree with the reviewer that the parameters are not disconnected from each other and we do agree that understanding their relationships is of particular interest. However, finding a realistic scenario seems very complex, in particular since the study deals with five ECV in a coupled system continent-ocean-atmosphere, involving both thermodynamic and dynamic processes. Such coupled systems imply chicken-and-egg relationships between variables that might be difficult to emphasize and understand. Only sensitivity tests with forced and/or coupled models, supposed realistic, could shed the light on the relationships between the variables. As mentioned in the Summary and Conclusion section, a precise quantification of the mechanisms responsible for the observed variability will be dealt with in an anticipated companion paper, as a second step, based on model outputs (from the hydro-dynamical model SYMPHONIE, in which for instance for the SST, all terms of the mixed layer temperature budget are available on regular gridded points).

Moreover, we note that proposing a common mechanism between variable might be sometimes counter-intuitive. We discussed the case of the inverse response of SST and SLA to ENSO in the SCS (lines 658-663).

*My another main criticism is regarding precipitation and river discharge, as the authors just include Vietnamese data. Why contributions from China, which limits SCS to a huge extent, are not included at all? Not a single word is mentioning this, and I do think Pearl River might have a significant river discharge into SCS?*

We understand precipitation changes in the southern part of China could have been analyzed as well. Because the ms. is quite long, we decided to put the emphasis on Vietnam in situ stations only, and Vietnam is the first author's affiliation. The Pearl River mouth is located in East China Sea, not in the South China Sea. We understand its fresh water flow likely influences the sea surface salinity distribution away from the ECS in the SCS. While sea surface salinity is also an ECV, it is not analyzed in the present ms.

*I have noted a number of additional minor points, but since I'm not recommending publication at this stage it is not worth listing them now, as the authors need to rewrite the entire manuscript in a short, precise way omitting all redundancies, but highlighting their results instead.*

We will be happy to have the list of minor points, assuming they are still relevant for our revised ms., and are confident these will further help us to improve our ms.

*I'm sorry I cannot be more positive at this moment.*

Thanks again for your comments, anyway.